# Overcoming therapeutic resistance in oncolytic herpes virotherapy by targeting IGF2BP3-induced NETosis in malignant glioma

Weiwei Dai[1,2], Ruotong Tian[1], Liubing Yu[1,2], Shasha Bian [1,2], Yuling Chen[1], Bowen Yin[1,2], Yuxuan Luan[1,2], Siqi Chen[1,2], Zhuoyang Fan[1,3], Rucheng Yan[1], Xin Pan [4,5], Yingyong Hou[6], Rong Li[7], Juxiang Chen[7] ✉ & Minfeng Shu [1,2] ✉

Oncolytic virotherapy holds promise for cancer treatment, but the factors determining its oncolytic activity remain unclear. Neutrophil extracellular traps (NETs) are associated with cancer progression, yet their formation mechanism and role in oncolytic virotherapy remain elusive. In this study, we demonstrate that, in glioma, upregulation of IGF2BP3 enhances the expression of E3 ubiquitin protein ligase MIB1, promoting FTO degradation via the ubiquitin-proteasome pathway. This results in increased m6A-mediated CSF3 release and NET formation. Oncolytic herpes simplex virus (oHSV) stimulates IGF2BP3-induced NET formation in malignant glioma. In glioma models in female mice, a BET inhibitor enhances the oncolytic activity of oHSV by impeding IGF2BP3-induced NETosis, reinforcing virus replication through BRD4 recruitment with the CDK9/RPB-1 complex to HSV gene promoters. Our findings unveil the regulation of m6A-mediated NET formation, highlight oncolytic virus-induced NETosis as a critical checkpoint hindering oncolytic potential, and propose targeting NETosis as a strategy to overcome resistance in oncolytic virotherapy.

Glioblastoma (GBM), the most common and deadliest primary brain malignancy in adults[1], consists of a diverse array of neoplastic and non-neoplastic cell types that collectively form the unique tumor micro-environment (TME) within the brain[2]. Non-neoplastic cells within the TME establish intricate reciprocal interactions with neoplastic cells, promoting tumor growth and invasion, and influencing the response to standard-of-care and emerging immunotherapies[3]. The most abundant cell type within the glioma TME are tumor-associated

myeloid cells (TAMs), which are of hematopoietic origin and include monocytes/monocyte-derived macrophages and neutrophils[4].

NETosis is a type of inflammatory cell death in neutrophils characterized by the formation of neutrophil extracellular traps (NETs)[5], consisting of DNA and proteins, such as neutrophil elastase (NE) and histones. NETs have been observed in various types of tumors[6,7]. It is demonstrated that molecules associated with NETs, such as high mobility group protein B1 (HMGB1), NE, and MMP9, can induce the

[1]Department of Pharmacology, School of Basic Medical Sciences, Shanghai Medical College, Fudan University, Shanghai, China. [2]Key Laboratory of Medical Molecular Virology (MOE/NHC/CAMS), Shanghai Frontiers Science Center of Pathogenic Microorganisms and Infection, School of Basic Medical Sciences, Shanghai Medical College, Fudan University, Shanghai, China. [3]Department of Interventional Radiology, Zhongshan hospital, Fudan University, Shanghai, China. [4]School of Basic Medical Sciences, Fudan University, Shanghai, China. [5]Nanhu Laboratory, National Center of Biomedical Analysis, Beijing, China. [6]Department of Pathology, Zhongshan hospital, Fudan University, Shanghai, China. [7]Department of Neurosurgery, Shanghai Changhai Hospital, Naval Medical University, Shanghai, China. ✉e-mail: juxiangchen@smmu.edu.cn; minfeng_shu@fudan.edu.cn

proliferation and invasion of cancer cells, promote extracellular matrix remodeling, and modulate other inflammatory cells[6]. Although a few studies have described a role for NETs in immunotherapy response[8], the mechanisms by which NETosis occurs and the implications of NETosis for tumor immunotherapies are not yet fully understood.

The N6-methyladenosine (m6A) RNA modification is prevalent in eukaryotic mRNAs and is added by the m6A methyltransferase complex (MTC), comprising METTL3, METTL14, WTAP, VIRMA, and RBM15. FTO and ALKBH5 are m6A demethylases that aid in the removal of this modification. m6A-modified mRNAs are recognized by specific reader proteins[9]. Insulin-like growth factor 2 mRNA-binding proteins (IGF2BPs), comprising IGF2BP1/2/3, are reader proteins that regulate mRNA fate[10]. IGF2BP3, a carcinoembryonic protein highly expressed during embryonic development, is upregulated in various cancers, including GBM[11]. It is not yet clear how IGF2BP3 affects the immune microenvironment and whether it influences the response of gliomas to immunotherapy. Furthermore, the inherent regulatory relationship between m6A-associated proteins, including readers and erasers, requires further clarification.

Oncolytic virotherapy is a well-established cancer treatment that utilizes virus-based therapies for direct application to tumors. Talimogene laherparepvec (T-VEC) is a genetically modified herpes simplex type 1 virus which incorporates a knock-in of granulocyte-macrophage colony-stimulating factor (GM-CSF) and a deletion of the neurovirulence factor infected cell protein 34.5 (ICP34.5). These modifications aim to improve the virus's replication in tumor cells with interferon (IFN) signaling defects, which are frequently observed in many types of tumors[12,13]. Oncolytic viruses (OVs) directly lyse tumor cells, leading to the release of soluble antigens and danger signals, which drive antitumor immunity[14]. However, the OVs exhibit variable therapeutic effects among cancer patients, and resistance to OVs is a common challenge. Although most research has concentrated on the role and mechanism of immune activation induced by OVs, the mechanisms underlying the modulation of the TME by OVs are still not fully understood. Specifically, it is uncertain whether treatment with OVs results in concurrent immunosuppression of the TME.

In this work, we report that IGF2BP3 initiates a reciprocal regulation between MIB1 and FTO to drive m6A-mediated NET formation and glioma progression. oHSV treatment stimulates the IGF2BP3/MIB1/FTO axis to boost virus replication, and this mechanism concurrently brings NETosis, ultimately leading to diminished oncolytic activity. Our findings provide insights into an inherent regulatory mechanism between RNA m6A reader and eraser, and delineate an oncolytic virus-induced NETosis effect. Targeting NETosis could be a promising strategy to overcome resistance in oncolytic virotherapy.

## Results
### IGF2BP3 facilitates NETosis and glioma survival
Using single-sample gene set enrichment analysis (ssGSEA), we found a significant positive correlation between IGF2BP3 expression and infiltrating neutrophils in gliomas (Supplementary Fig. 1a, b), suggesting a possible role of IGF2BP3 in regulating neutrophil function.

To investigate the specific impact of IGF2BP3 on neutrophil function, we established three in vitro co-culture models of glioma cells with neutrophils from distinct sources: tumor-associated neutrophils (TANs) from glioma tissue in mice with in situ tumorigenesis (Fig. 1a), bone marrow neutrophils from healthy mice (labeled as "neutrophil"; Supplementary Fig. 1c), and human HL-60 cells induced to exhibit neutrophil-like characteristics (called "induced neutrophils"; Supplementary Fig. 1d). Immunofluorescence staining was performed to detect specific markers for neutrophils (myeloperoxidase, MPO) and NETosis (citrullinated histone H3, H3cit)[15]. We found that co-culturing glioma cells with neutrophils induced NETosis, which was inhibited by DNase I and enhanced by PMA treatment (Fig. 1b, c). Interestingly, the effect was more pronounced in tumor cells

overexpressing IGF2BP3. This trend was also observed in bone marrow neutrophils (Supplementary Fig. 1e, f) and HL-60 cells (Supplementary Fig. 1g, h). However, IGF2BP3 failed to induce NETosis in neutrophils isolated from mice lacking the enzyme PAD4(PAD4$^{-/-}$), which is essential for NET formation[16] (Supplementary Fig. 1e, f). Electron microscopy analysis confirmed the IGF2BP3-induced NETosis, characterized by the formation of membrane protrusions and fibers (NETs) (Fig. 1d and Supplementary Fig. 1i). In the contacted co-culture assay, IGF2BP3 overexpression alone led to an increase in the number of glioma cells, which is consistent with a previous study[17]. Remarkably, IGF2BP3-induced NETosis significantly enhanced glioma cell survival in an IGF2BP3-dependent manner (Fig. 1e and Supplementary Fig. 1j, k), and this effect was abolished in neutrophils from PAD4$^{-/-}$ mice (Supplementary Fig. 1l). These findings indicate that IGF2BP3 enhances glioma cell survival both through NET-dependent and NET-independent mechanisms. Furthermore, the basal level of NET formation in TANs is higher than that in control neutrophils (from bone marrow) (Supplementary Fig. 1m, n). However, there is no statistically significant difference between the glioma cell growth induced by TANs and that induced by control neutrophils (Supplementary Fig. 1o). One possible explanation is that, in the contacted co-culture system, glioma cells expressing a high basal level of IGF2BP3 can stimulate a relatively strong NET formation. This could potentially overshadow the differences in basal NET formation between TANs and the control neutrophils.

Consistent with the in vitro experiments, IGF2BP3 down-regulation reduced glioma survival in wild-type mice, as evidenced by decreased tumor size (Fig. 1f–h), increased survival time (Fig. 1i), and decreased neutrophil infiltration (CD66b +, MPO) and NETosis (H3cit +) (Fig. 1j–m). Conversely, IGF2BP3 overexpression had the opposite effect (Fig. 1f, g, j–o). However, the effect was significantly reduced in PAD4$^{-/-}$ mice (Fig. 1f, g, j–o). The protein levels resulting from the overexpression or downregulation of IGF2BP3 in the GL261 cells used in the study were also demonstrated (Fig. 1p). These results suggest that NETosis plays a partial role in the promotion of tumor survival by IGF2BP3.

### CSF3 secretion mediates IGF2BP3-induced NETosis and tumor survival
To understand how IGF2BP3 promotes NETosis, we conducted mRNA sequencing (mRNA-seq) and found that the chemokine signaling pathways, including neutrophil chemotaxis, were significantly enriched (Fig. 2a, b). CSF3, also named G-CSF[18], a regulator of neutrophil recruitment, was upregulated in IGF2BP3-transfected cells (Fig. 2d, e) and downregulated in IGF2BP3-knockdown cells (Fig. 2f, g), consistent with the RNA-seq data (Fig. 2c). ELISA analysis confirmed IGF2BP3-induced secretion of CSF3 (Fig. 2h, i). We investigated whether IGF2BP3-induced NETosis is dependent on CSF3 and found that the use of CSF3 recombinant protein significantly increased the IGF2BP3-induced NETosis of neutrophils from different origins. This effect was reversed by the neutralizing antibody to CSF3 (Fig. 2j-p). We also observed that the promotion of glioma cell survival by IGF2BP3-induced NETosis in a contacted co-culture assay could be blocked by CSF3 neutralizing antibodies and enhanced by CSF3 recombinant protein (Fig. 2q–t). Notably, these effects were barely detectable in the co-culture model of neutrophils with PAD4-knockout (Fig. 2k, o, s, t). These findings suggest that CSF3 secretion mediates IGF2BP3-induced NETosis and tumor survival.

### IGF2BP3 upregulates CSF3 expression mediated by FTO
We investigated how IGF2BP3 upregulates CSF3 expression and found that it significantly increases overall m6A levels in cells, as confirmed by m6A dot-blot and liquid chromatography mass spectrometry (Fig. 3a, b). IGF2BP3 downregulated the expression of both m6A-associated writers (METTL3, METTL14) and erasers (FTO,

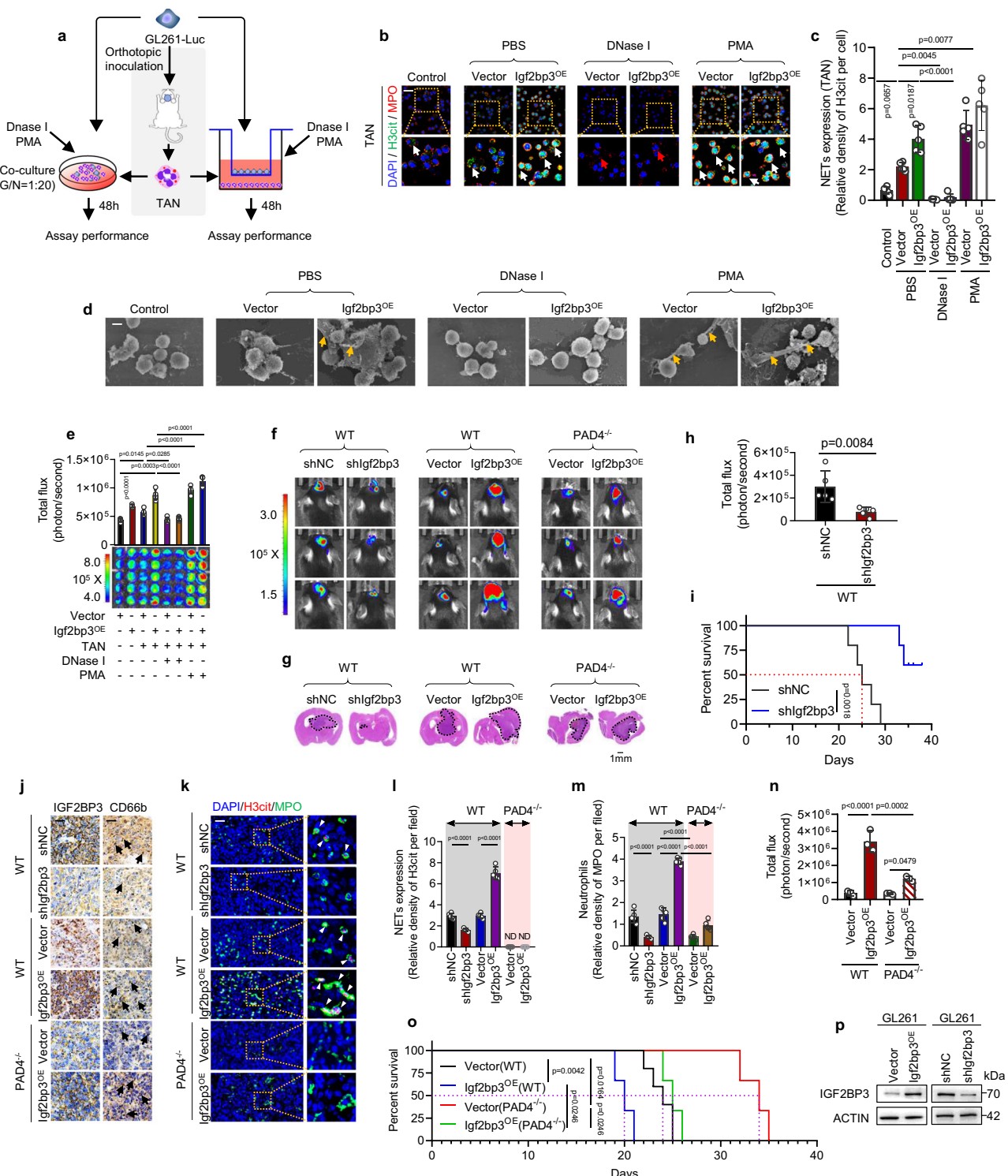

ALKBH5), with a specific focus on FTO due to its more pronounced effect in both U87MG and GL261 glioma cell lines (Fig. 3c). IGF2BP3 reduced the mRNA expression levels of METTL3, METTL14, and ALKBH5, but not FTO (Fig. 3d). RIP assay unveiled that IGF2BP3 failed to bind to the mRNA of FTO (Fig. 3e). Cycloheximide chase assays showed that IGF2BP3 facilitated FTO protein degradation (Fig. 3f, g). The proteasome inhibitor MG132, but not the lysosome inhibitor chloroquine, significantly blocked the IGF2BP3-induced FTO degradation (Fig. 3h-j). Further mapping analysis identified that the lysine-48 (K48) polyubiquitin mediated the IGF2BP3-induced FTO degradation (Fig. 3k).

Based on our findings, we propose that IGF2BP3 upregulates CSF3 expression through m6A modification. Knockdown of FTO increased both mRNA and protein levels of CSF3, while overexpression had the opposite effect (Fig. 3l–n). FTO was found to promote significant degradation of CSF3 mRNA (Fig. 3o, p), and the SRAMP web tool (http://www.cuilab.cn /sramp)[19] predicted potential m6A modification sites on the 3′UTR of CSF3 mRNA (Fig. 3q). MeRIP assay confirmed that knockdown of FTO increased m6A levels of CSF3 mRNA (Fig. 3r). Moreover, we generated CSF3-MUT/WT luciferase reporters (Fig. 3s) to show that FTO knockdown only increased the luciferase activity of CSF3-WT, not CSF3-MUT (Fig. 3t). IGF2BP3 was also found to stabilize

**Fig. 1 | IGF2BP3 facilitates NETosis and glioma survival in vitro and in vivo.**
**a** Schematic representation of the contacted co-culture assay and uncontacted transwell assay. "GL261-Luc" indicates GL261 cells expressing luciferase, and "TAN" indicates tumor-associated neutrophils isolated from the GL261-derived in-situ glioma model. "G/N = 1:20" represents the ratio of glioma cells to neutrophils.
**b**, **c** Representative immunofluorescence images **b** and quantification **c** of NET formation in the GL261 cell/TAN uncontacted transwell assay. "Control" indicates TANs unstimulated with GL261 glioma cells in vitro. White arrows indicate NETs co-stained with H3cit, MPO, and DAPI, while red arrows indicate intact TAN. Scale bars, 20 μm. Representative images of $n = 5$. $n = 5$ biological independent samples. **d** Electron microscopic analysis of TAN co-cultured with GL261-Luc cells. Yellow arrows indicate NETs. Scale bars, 5 μm. Representative images of $n = 3$.
**e** The GL261 cell survival induced by NETs in the contacted co-culture assay was detected. $n = 5$ biological independent samples. **f**, **h**, **n** Representative images **f** of the luciferase signal from C57BL/6 mice. Tumor size was estimated by quantifying luciferase activity ($n = 5$ mice per group for **h**, $n = 3$ mice per group for **n**). **g** Hematoxylin and eosin (HE) staining analysis of the tumor volume. The tumor size was labeled with a dashed line. **i**, **o** Survival curves ($n = 5$ mice per group). **j** The expression of IGF2BP3 and neutrophil infiltration (CD66b, labeled with black arrows) in the indicated tumor tissues. Scale bars, 20 μm. Representative images of $n = 3$. **k**–**m** Representative immunofluorescence images **k** and quantification **l** of NET formation and infiltrated neutrophil **m** in the indicated tumor tissues. White arrows indicate NETs co-stained with H3cit, MPO, and DAPI. Scale bars, 20 μm. $n = 5$ biological independent samples. **p** The protein level of IGF2BP3 overexpression/downregulation in the glioma cells used in Fig. 1f. $n = 3$ biologically independent experiments. Statistical significance was determined using one-way ANOVA in **c**, **e**, **l**–**n**, or two-tailed Student's t-test in **h**, or the log-rank (Mantel-Cox) test in **o**. Data represent the mean ± SD. Source data are provided in the Source Data file.

CSF3 mRNA (Fig. 3u) and interact with CSF3 (Fig. 3v, w), as confirmed by RIP-qPCR and dual luciferase reporter assays. These results suggest that IGF2BP3 upregulates CSF3 expression through an m6A-dependent mechanism mediated by FTO.

### Identifying MIB1-mediated degradation of FTO protein
To unravel the mechanism by which IGF2BP3 promotes proteasomal degradation of FTO, we performed dual-omics analysis of IGF2BP3 RNA sequencing and FTO mass spectrometry. Our results reveal that three E3 ubiquitin ligases, namely MIB1, HERC2, and HUWE1, are enriched by both transcriptomic and proteomic analyses (Fig. 4a, b). Immunoprecipitation assays confirmed that FTO interacts with MIB1, but not HERC2 and HUWE1 (Fig. 4c–f). Mapping analysis identified that the N-terminal of MIB1 (1-429aa) interacts with the C-terminal of FTO (327-505aa) (Fig. 4g–j). MIB1 could downregulate FTO protein levels in a dose-dependent manner (Fig. 4l), while not affecting its mRNA (Fig. 4k). Moreover, overexpression of MIB1 resulted in an increase in FTO polyubiquitination (Fig. 4m). Our analysis predicted eight potential ubiquitination sites in FTO protein, and we found that lysine-216 (K216) residue mediated FTO ubiquitination (Fig. 4n, o, p). Importantly, we confirmed that lysine-216 mutant (K216R) significantly blocked MIB1-induced FTO ubiquitination (Fig. 4q). In conclusion, we propose that MIB1 interacts with FTO and mediates its protein degradation via ubiquitination at lysine-216 (K216).

### IGF2BP3/MIB1/FTO forms a positive feedback loop enhancing NETosis
We investigated how IGF2BP3 upregulates MIB1 expression and found that overexpression of IGF2BP3 increased both the mRNA and protein levels of MIB1, and vice versa (Fig. 5a, b). We also found that IGF2BP3 attenuated the degradation of MIB1 mRNA (Fig. 5c) and confirmed the interaction between MIB1 and IGF2BP3(Fig. 5d). Using the SRAMP web tool (http://www.cuilab.cn/sramp), we predicted potential m6A modification sites on the coding sequence of MIB1 mRNA (Fig. 5e) and found that IGF2BP3 upregulated MIB1 expression through an m6A-dependent mechanism (Fig. 5f–h). In addition, knockdown of FTO increased both the mRNA and protein levels of MIB1, while overexpression of FTO decreased both mRNA and protein levels of MIB1(Fig. 5i, j). RNA decay assay showed that knockdown of FTO stabilized MIB1 mRNA (Fig. 5k). MeRIP and dual luciferase reporter assays revealed that FTO mediated an m6A-dependent mechanism of MIB1 expression (Fig. 5l, m). These findings suggest that IGF2BP3 upregulates MIB1 expression via an m6A-dependent mechanism mediated by FTO and indicate a reciprocal regulation between MIB1 and FTO.

To investigate the functional association of the IGF2BP3/MIB1/FTO loop with IGF2BP3-induced NETosis and tumor survival, we disrupted the pathway using a genetic approach. Overexpression of FTO effectively suppressed the IGF2BP3-induced upregulation of MIB1 and overall increase in m6A (Fig. 5n, o), inhibiting IGF2BP3-induced NETosis and glioma cell survival (Fig. 5p–r). Overexpression of MIB1 also effectively rescued the upregulation of FTO and reduction of total m6A caused by IGF2BP3 knockdown (Fig. 5s, t). Additionally, IGF2BP3 knockdown-induced NETosis suppression (Fig. 5u, v) and glioma cell survival was rescued by MIB1 overexpression (Fig. 5w). These results were further verified in bone marrow neutrophils (Supplementary Fig. 2a–f) and human HL-60 cells (Supplementary Fig. 2g–l). Notably, the FTO-216R mutant, which is resistant to ubiquitination by MIB1, markedly inhibited the IGF2BP3-induced upregulation of CSF3 and the overall m6A level (Supplementary Fig. 2m–r). We further revealed that the IGF2BP3/MIB1/FTO pathway forms a positive feedback loop in both T98G cells and primary cultured human glioma cells (Supplementary Fig. 3a–x). These findings suggest that the IGF2BP3/MIB1/FTO pathway establishes a positive feedback loop that enhances overall m6A and NETosis, promoting glioma progression (Fig. 5x).

### oHSV enhances IGF2BP3-dependent NETosis and supports tumor survival
As multiple pathways enriched by IGF2BP3 overexpression are related to virus infection (Fig. 2a), we conducted KEGG analysis of the mRNA-seq data and found that herpes simplex virus infection was the top enriched pathway (Fig. 6a). Moreover, analysis of mRNA-seq data from an oHSV infection model showed that pathways of neutrophil extracellular trap formation were enriched, indicating a potential association between HSV infection and IGF2BP3-induced NETosis (Fig. 6b). Although oHSV shows promise as a therapeutic agent for treating malignant glioma[20], its efficacy may vary among cancer patients. We found that GL261 cells were the most resistant to oHSV, while U87MG cells were the most sensitive among the four glioma cell lines (Fig. 6c; Supplementary Fig. 4a). To investigate the precise impact of oHSV infection on NETosis, we established an in vitro co-culture model of neutrophils with glioma cells infected with oHSV (Fig. 6d). The co-culture supernatant of oHSV-infected glioma cells with neutrophils resulted in NETosis production, which could be inhibited by DNase I and enhanced by PMA treatment (Fig. 6e, f). Notably, this effect was more pronounced in tumor cells infected with oHSV. We also observed oHSV-induced NETosis in bone marrow neutrophils derived from wild-type rather than PAD4$^{-/-}$ mice (Supplementary Fig. 4b, c). Furthermore, oHSV-induced NETosis in the co-culture system promoted glioma cell survival (Fig. 6g and Supplementary Fig. 4d).

In vivo, oHSV treatment reduced glioma survival and tumor size in wild-type mice (Fig. 6h–j), with even greater effects seen in PAD4$^{-/-}$ mice. Moreover, oHSV treatment in wild-type mice increased NETosis (H3cit + ) (Fig. 6k, l), but this effect was significantly reduced in PAD4$^{-/-}$ mice. These results suggest that oHSV-induced NETosis plays a critical role in reducing the oncolytic activity of oHSV. Viral protein ICP0 was also detected (Fig. 6m).

To investigate how oHSV promotes NETosis, we found that oHSV infection increased IGF2BP3 and MIB1 expression while suppressing

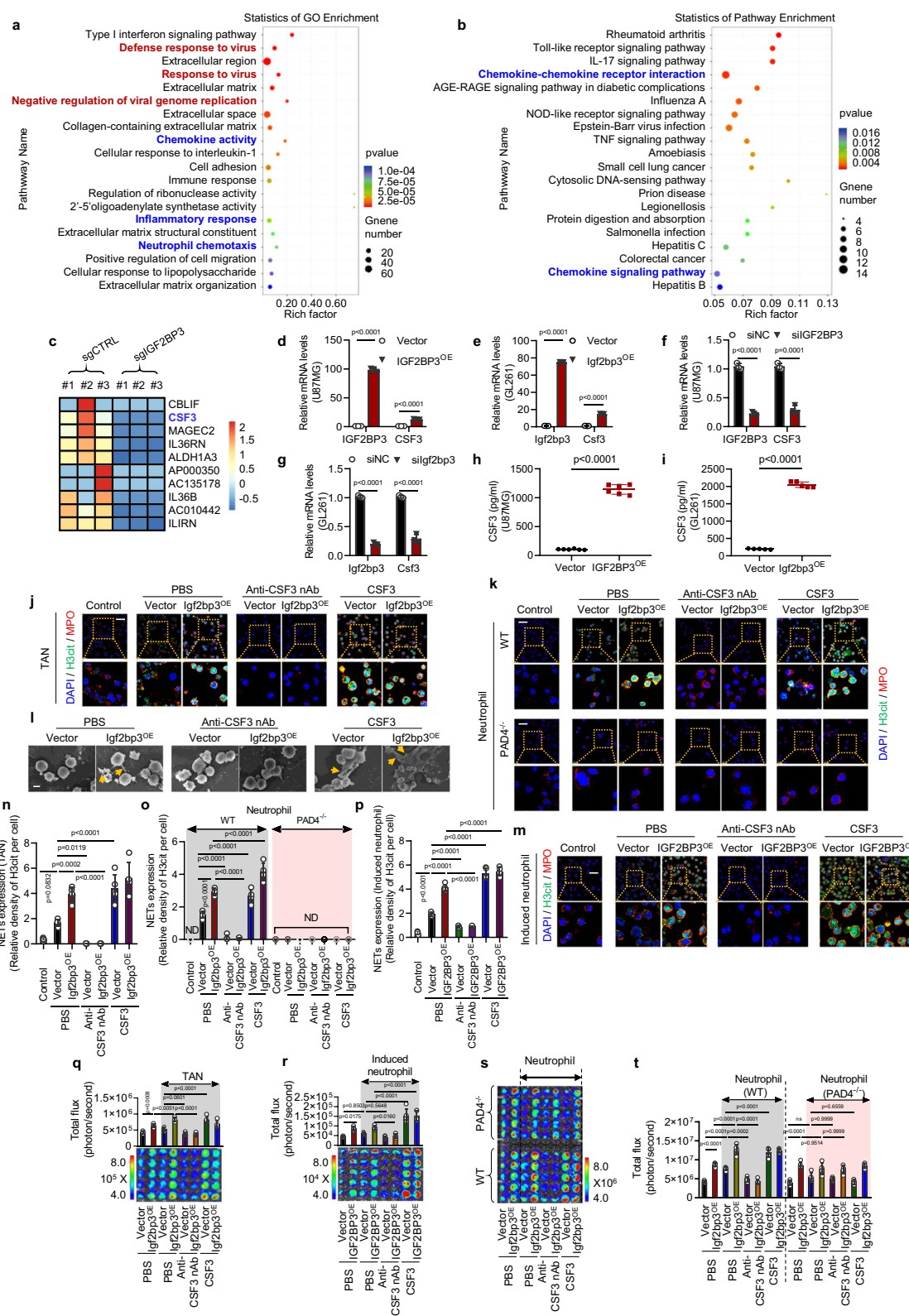

FTO in a time-dependent manner, along with the detection of viral proteins ICP0, ICP8, and gC (Fig. 6n, o). Additionally, oHSV infection upregulated CSF3 (Csf3) mRNA levels in both oHSV-sensitive U87MG and oHSV-resistant GL261 glioma cells (Fig. 6p, q). Our data suggest that oHSV triggers the IGF2BP3/MIB1/FTO feedback loop, and oHSV-induced NETosis can be blocked by Igf2bp3 knockdown (IGF2BP3 knockout) and CSF3 neutralizing antibodies (Fig. 6r, s and Supplementary Fig. 4e–i). Furthermore, oHSV-induced NETosis in the

contacted co-culture system promoted glioma cell survival in an IGF2BP3/CSF3-dependent manner (Fig. 6t, u and Supplementary Fig. 4j–m).

To eliminate the possibility of oHSV-induced NETosis resulting from direct oHSV virus infection, we tested for viral proteins in the supernatant from co-culture systems of oHSV-infected tumor cells at specified time points (Supplementary Fig. 4n–p). No viral proteins were detected, which suggests that oHSV-induced NETosis is not

**Fig. 2 | Inhibiting the IGF2BP3-induced secretion of CSF3 suppresses NETosis and reduces glioma survival. a, b** mRNA-seq analysis was conducted on U87MG cells with IGF2BP3 overexpression **a** or knockout **b**, together with the corresponding control groups. The top 20 pathways with the smallest P values were selected for GO enrichment analysis. $n = 3$ biological independent experiments. **c** The heatmap displays the top 10 genes that were affected by IGF2BP3 knockout, based on mRNA-seq results. $n = 3$ biological independent experiments. **d–g** RT-qPCR analysis revealed the relative levels of CSF3(Csf3) mRNA in U87MG and GL261 cells with IGF2BP3 knockdown or overexpression. $n = 3$ biological independent experiments. **h, i** ELISA analysis demonstrated the relative levels of CSF3 protein in U87MG and GL261 cells. $n = 6$ biological independent samples for panel **h**, $n = 5$ biological independent samples for panel **i**. **j, n** Representative immunofluorescence images **j** and quantification **n** of NET formation in the GL261 cell/TAN uncontacted transwell assay treated with CSF3 neutralizing antibody (Anti-CSF3 nAb) or recombinant protein (CSF3). "Control" indicates TANs unstimulated with glioma cells in vitro. Scale bars, 20 μm. Representative images of $n = 5$. **k, m, o, p** Representative immunofluorescence images **k, m** and quantification **o, p** of NET formation in the GL261 cell/neutrophil and U87MG cell/induced neutrophil (HL-60) uncontacted transwell assays treated with Anti-CSF3 nAb or CSF3 recombinant protein. "Control" indicates neutrophils unstimulated with glioma cells in vitro. "ND" as undetected. Scale bars, 20 μm. Representative images of $n = 5$. **l** Electron microscopic analysis of TANs co-cultured with GL261-Luc cells stably expressing vector or Igf2bp3 with the indicated treatment. Yellow arrows indicate NETs, characterized by membrane protrusions and fiber formation. Scale bars are 5 μm. Representative images of $n = 3$. **q–t** The effect of CSF3 neutralizing antibody and recombinant protein on NET-induced tumor cell survival in contacted co-culture assays was determined using a luciferase assay. $n = 5$ biological independent samples. Statistical significance was determined using two-tailed Student's t-test in **d–i**, or one-way ANOVA in **n–r, t**. Data represent the mean ± SD. Source data are provided in the Source Data file.

triggered by direct oHSV virus infection. Importantly, it was further confirmed that oHSV enhances NETosis in an IGF2BP3-dependent manner in primary cultured tumor cells and neutrophils obtained from glioma patients (Supplementary Fig. 5a–l).

## Inhibition of IGF2BP3 enhances oncolytic activity of oHSV in glioma

Prior studies have suggested that the small molecule JQ1 can suppress IGF2BP3 expression in neonatal megakaryocytes[21,22]. Indeed, our study demonstrated that BET inhibitors (JQ1, I-BET151) time-dependently inhibited the expression of both IGF2BP3 and CSF3 in glioma cells (Supplementary Fig. 6a–f). Treatment with BET inhibitors also significantly reduced overall m6A levels (Supplementary Fig. 6g, h). Our results further indicated that enforced IGF2BP3 expression significantly reduced the ability of BET inhibitors to reduce tumor survival (Supplementary Fig. 6i–k), extend survival time (Supplementary Fig. 6l), and inhibit NETosis (Supplementary Fig. 6m, n). IGF2BP3 expression in tumor sections was also detected (Supplementary Fig. 6o). Addition of exogenous CSF3 significantly increased NETosis and diminished the effectiveness of BET inhibition in reducing NETosis (Supplementary Fig. 6p, q, r, s) and inhibiting tumor survival (Supplementary Fig. 6t, u) in both TAN and induced neutrophil models. Collectively, these findings provide strong evidence that the effects of BET inhibitors on NETosis and tumor survival inhibition result, in part, from the downregulation of IGF2BP3 and subsequent decrease in CSF3 expression. Additionally, JQ1 treatment can significantly reduce oHSV-induced NETosis and tumor survival in vitro (Fig. 7a–d; Supplementary Fig. 7a–c).

To investigate whether JQ1 can enhance the antitumor activity of oHSV in vivo, we established a subcutaneous glioma model (Fig. 7e) and found that JQ1 markedly enhanced the oncolytic activity of both oHSV and oHSV-anti-PD1 viruses, resulting in reduced tumor weight and volume (Fig. 7f–h). Additionally, JQ1 treatment enhanced the expression of all classes of HSV viral genes (Fig. 7i). These findings were further validated in an in-situ glioma model (Fig. 7j), as evidenced by prolonged survival time (Fig. 7k), reduced tumor size (Fig. 7l–n), and decreased neutrophil infiltration (CD66b + ) and NETosis (H3cit + ) (Fig. 7o–q). JQ1 treatment also effectively hindered the oHSV-induced increase in overall m6A levels (Supplementary Fig. 7d), as well as the upregulation of IGF2BP3 and CSF3 (Supplementary Fig. 7e).

Additionally, overexpression of IGF2BP3 significantly enhanced the expression of all classes of HSV viral genes at both mRNA and protein levels (Supplementary Fig. 7f–i). Conversely, this effect was reversed by IGF2BP3 knockdown (Supplementary Fig. 7j–m), indicating that virus-stimulated IGF2BP3 can promote self-replication. JQ1 increased viral gene expression in vitro and in vivo (Fig. 7i, o; Supplementary Fig. 7n–p). Our study further showed that oHSV infection induced the recruitment of BRD4 and Rpb1 to the promoters of HSV viral genes in glioma cells (Supplementary Fig. 7q, r). Treatment with

JQ1 significantly enhanced the interaction between BRD4 and the CDK9/Rpb1 complex (Supplementary Fig. 7s). Furthermore, JQ1 reinforced the oHSV-induced recruitment of BRD4 to the promoters of HSV viral genes (Supplementary Fig. 7t) and reduced the expression of INF-beta induced by oHSV infection (Supplementary Fig. 7u). These results suggest that JQ1 promotes oHSV replication by enhancing viral gene transcription through the BRD4/CDK9/Rpb1 complex.

## IGF2BP3-induced NET is linked to poor prognosis in glioma patients

IGF2BP3 was significantly overexpressed in gliomas, correlating with higher glioma grade and poor prognosis in patients (Supplementary Fig. 8a–e). Its expression was also elevated in human glioma cells and glioma stem cells (GSCs) (Supplementary Fig. 8f). Additionally, IGF2BP3 showed a positive correlation with CD66b, MPO, and H3cit expression in glioma patients (Supplementary Fig. 8g–i), indicating its potential role in NETosis induction. Furthermore, CD66b, MPO, and H3cit expression levels were positively associated with glioma grade and poor prognosis in patients (Supplementary Fig. 8d, j–m).

## Discussion

Our research shows that oHSV boosts IGF2BP3, enhancing its replication. However, this triggers an overall rise in m6A levels in infected tumor cells via the IGF2BP3/MIB1/FTO feedback loop, ultimately leading to CSF3-mediated NETosis. The BET inhibitor acts as an oHSV enhancer, reducing IGF2BP3-induced NETosis while promoting virus replication simultaneously (Fig. 8). Despite the common belief that OVs-induced immune activation is a primary mechanism for their antitumor effects, our findings suggest that the NETosis induced by OVs serves as a critical checkpoint, hindering their oncolytic activity.

It is well-established that m6A-associated factors function in an orchestrated manner[23]. Nevertheless, we demonstrate that the m6A reader protein IGF2BP3 enhances overall m6A levels, which are typically controlled by writers or erasers. IGF2BP3 triggers a reciprocal regulation between MIB1 and FTO, resulting in an overall increase of m6A levels, which reveals an inherent regulatory pattern between reader and eraser. Given that IGF2BP3 also possesses m6A sites, we have shown that FTO cannot modulate the expression of IGF2BP3 across various cell lines (Supplementary Fig. 9a–d). Additionally, we observed that IGF2BP3 overexpression led to the suppression of the writer proteins METTL3 and METTL14. This may represent a finely-tuned regulatory response to maintain the balance of m6A levels, as suggested by the overall increase in total m6A levels. The coordinated regulation of both eraser and writer proteins supports the idea that cells strive to maintain m6A homeostasis. Moreover, we further revealed that METTL3 overexpressing had little effect on IGF2BP3 induced MIB1 expression and NETs formation (Supplementary Fig. 10a–f), suggesting that METTL3 does not play a major role in IGF2BP3-mediated NETosis.

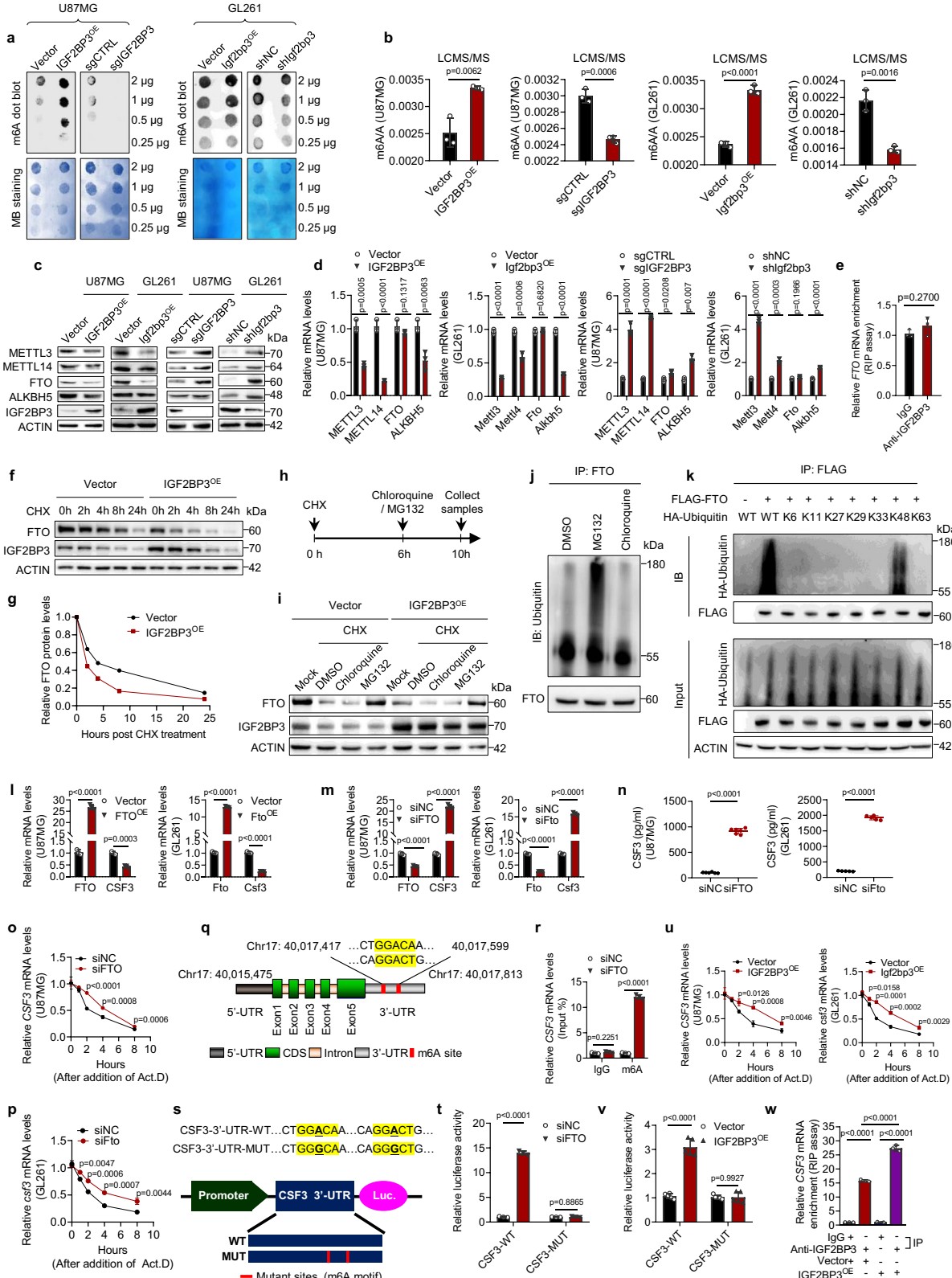

Neutrophils are commonly found in various types of cancers as tumor-associated neutrophils (TANs). However, the precise roles of neutrophils in cancer are still a matter of debate[24]. On one hand, neutrophils can induce tumor cell killing through the release of tumor necrosis factor-related apoptosis-inducing ligand (TRAIL) or reactive oxygen species (ROS)[25,26]. On the other hand, neutrophils act as "accomplices" in promoting tumor progression. Notably, neutrophils

that infiltrate the brain have been found to be immunosuppressive due to their expression of arginase 1 (Arg-1) and programmed death-ligand 1 (PD-L1), and bone marrow-derived neutrophils pre-treated with 4T1 conditioned medium can suppress T cell proliferation ex vivo[27]. Mousset et al. illustrate that chemotherapy-induced inflammation confers chemoresistance by facilitating NETosis in malignant tumors, highlighting a therapeutic opportunity to target inflammatory NETs in

**Fig. 3 | IGF2BP3 promotes the degradation of FTO, resulting in increased total m6A levels and stabilization of CSF3 mRNA through an m6A-dependent mechanism.** a–d The impact of IGF2BP3 on total m6A levels a, b and m6A-related enzymes c, d. $n = 3$ biological independent experiments. e The interaction of IGF2BP3 with FTO mRNA. $n = 3$ biological independent experiments. f, g The impact of IGF2BP3 on FTO protein stability in U87MG cells. Quantification of FTO protein levels is shown g. $n = 3$ biological independent experiments. h, i The effect Chloroquine (50 μM) and MG132 (10 mM) on IGF2BP3-induced FTO protein stabilization in U87MG cells. The experimental design is shown h. $n = 3$ biological independent experiments for i. j, k The impact of chloroquine and MG132 treatments j as well as transfection with indicated HA-Ubiquitin mutants k on polyubiquitination modification of FTO protein in U87MG cells. $n = 3$ biological independent experiments. l–n The effect of FTO (Fto) on relative mRNA l, m and protein n levels of CSF3 (Csf3). $n = 3$ biological independent experiments for l, m. $n = 6$ or 5 biological

independent samples for n. o, p, u The impact of FTO (Fto) knockdown o, p or IGF2BP3 overexpression u on the CSF3 (Csf3) mRNA stability. $n = 3$ biological independent experiments. q A schematic diagram of CSF3 mRNA with the predicted 'm6A' sites with the highest confidence at the 3'UTR highlighted in red is shown. r The m6A modification of CSF3 in U87MG cells. The enrichment of m6A was calculated by m6A-IP/input and IgG-IP/input. $n = 3$ biological independent experiments. s A schematic diagram of the CSF3 luciferase reporter. t, v The relative activity of the WT or MUT luciferase reporters, based on the psiCHECK2 plasmid was determined in U87MG cells. $n = 3$ biological independent experiments for t. $n = 5$ biological independent experiments for v. w The interaction between IGF2BP3 protein and CSF3 mRNA in U87MG cells. $n = 3$ biological independent experiments. Statistical significance was determined using two-tailed Student's t-test in b, d, e, l–p, r, t, u, v, or one-way ANOVA in w. Data represent the mean ± SD. Source data are provided in the Source Data file.

cancer treatment[28]. In addition, the overexpression of IGF2BP3 alone can increase the number of glioma cells (Fig. 1e), indicating that IGF2BP3 induces cancer cell survival both in a NET-dependent and a non-NET-dependent manner. As a result, we conclude that the overexpression of IGF2BP3 in glioma cells triggered NET formation, which contributes to the progression of glioma. Blocking NETosis enhances the oncolytic activity of oHSV, highlighting the role of immunosuppressive neutrophils in the pathogenesis and therapeutic resistance of glioma treatment.

OVs treatment can modify the TME by promoting increased immune cell infiltration within tumors, thereby enhancing the effectiveness of immune checkpoint inhibitor (ICI) therapy and bolstering anti-cancer activity[29]. Unfortunately, the phase III MASTERKEY-265 trial has shown no significant improvement in progression-free survival or overall survival with the addition of talimogene laherparepvec (T-VEC) to pembrolizumab in patients with advanced melanoma[30]. Consistently, our data revealed no benefit in suppressing glioma survival with oHSV-anti-PD1 treatment compared to oHSV treatment. However, incorporating JQ1 significantly enhanced the anti-tumor activity of oHSV-anti-PD1 compared to oHSV treatment (Fig. 7m), suggesting that the TME remodeling induced by OVs may be more complex than currently understood. We observed a significant reduction in neutrophil presence following JQ1 plus oHSV treatment (Fig. 7p). Our results demonstrate that IGF2BP3 overexpression promotes neutrophil infiltration. Additionally, we have shown that JQ1 can decrease the expression of IGF2BP3. Consequently, it is plausible that the downregulation of IGF2BP3 induced by JQ1 contributes to the reduced presence of neutrophils observed in the JQ1 plus oHSV treatment group. Furthermore, MIB1 plays a critical role in activating the NOTCH signaling pathway by ubiquitinating and endocytosing NOTCH ligands[31]. NOTCH signaling is well-known for its involvement in numerous conserved signal transduction pathways related to immune cell development and homeostasis. Therefore, HSV-induced upregulation of MIB1 may also lead to the activation of the NOTCH signaling pathway, contributing to TME regulation. Furthermore, HSV-1-encoded miR-H16 directly targets and triggers the degradation of factor inhibiting HIF-1(FIH-1) which can interact with and inhibit the activity of MIB1[32], making it justifiable to presume that the HSV-1 miR-H16/FIH-1/MIB1 pathway may also contribute to HSV-1-induced reduction of FTO and promotion of NETosis. Recently, CSF3 has been implicated in the recruitment of immunosuppressive neutrophils to the brain, leading to a suppressive effect on T cells[27]. In fact, we demonstrate that oHSV infection can induce CSF3-mediated NETosis and promote tumor survival in malignant glioma, suggesting that the effect of NETs on oHSV resistance is independent on the regulation of antitumor immunity. Additionally, the OVs-stimulated immune activation not only helps combat tumors but also accelerates the clearance of the virus itself, creating a delicate balance between the two. Therefore, while boosting the OVs-induced immune activation may seem like a viable strategy, relieving the NETosis triggered by these

viruses may represent a more effective means of enhancing oncolytic activity.

Our study revealed that U87MG glioma cells exhibited the highest sensitivity to oHSV infection, as demonstrated in Fig. 6c. Several factors may account for this heightened sensitivity observed in U87MG cells compared to the other three glioma cell lines. Firstly, U87MG cells may express a higher level of cell surface receptors conducive to oHSV infection. This enhanced receptor expression might facilitate more efficient interactions between the virus and these cells, increasing their susceptibility to infection. Additionally, the immune status of U87MG cells may favor oHSV infection over the other cell lines. This immune advantage could manifest as a weaker immune response, including reduced antiviral defense mechanisms, thereby creating a more permissive environment for the virus to thrive and replicate. However, additional research is needed to delve into the potential mechanism.

It is shown that inhibition of m6A can attenuate HSV viral gene transcription[33]. Our results indicate that overexpression of IGF2BP3 can significantly enhance oHSV replication, suggesting that the upregulation of overall m6A levels induced by IGF2BP3 may be responsible for oHSV replication. Additionally, although CSF3 has been applied as a supportive therapy to prevent or treat chemotherapy-associated neutropenia in cancer patients, emerging investigations have drawn attention to the tumor-promoting effects of CSF3 on myeloid cells[34,35]. Our study demonstrated that CSF3 neutralizing antibody could significantly reduce oHSV-induced NETosis within the TME and tumor survival. This finding suggests that a combination of systemic administration of CSF3 and intratumoral delivery of oHSV armed with CSF3-neutralizing antibodies may represent a viable approach to enhance the anticancer activity of oHSV. Further investigation is warranted to explore the potential of this strategy.

The optimal strategies to enhance the oncolytic activity of OVs would theoretically not only strengthen the anti-tumor immune response, but also promote virus replication within malignancies. Our studies have shown that BET inhibitors meet the requirements by significantly reducing IGF2BP3-induced NETosis while promoting the recruitment of BRD4 with CDK9/RPB-1 complex to HSV gene promoters, thereby reinforcing virus replication. It is shown that BRD4 recruits the CDK9/Rpb1 complex to the promoter region to facilitate gene transcription[36], which is consistent with our findings. Additionally, BET inhibitors have been shown to penetrate the blood-brain barrier(BBB)[37], indicating that they are potential candidates of enhancer to facilitate the use of oHSV for glioma treatment.

In summary, our study revealed an epigenetic mechanism underlying NETosis during glioma progression, as well as the feature of oncolytic virus-induced NETosis. Further exploration of the role and mechanisms of NETosis could not only expand our current understanding of the complex microenvironmental landscape, but also pave the way for new therapeutic approaches for this challenging disease.

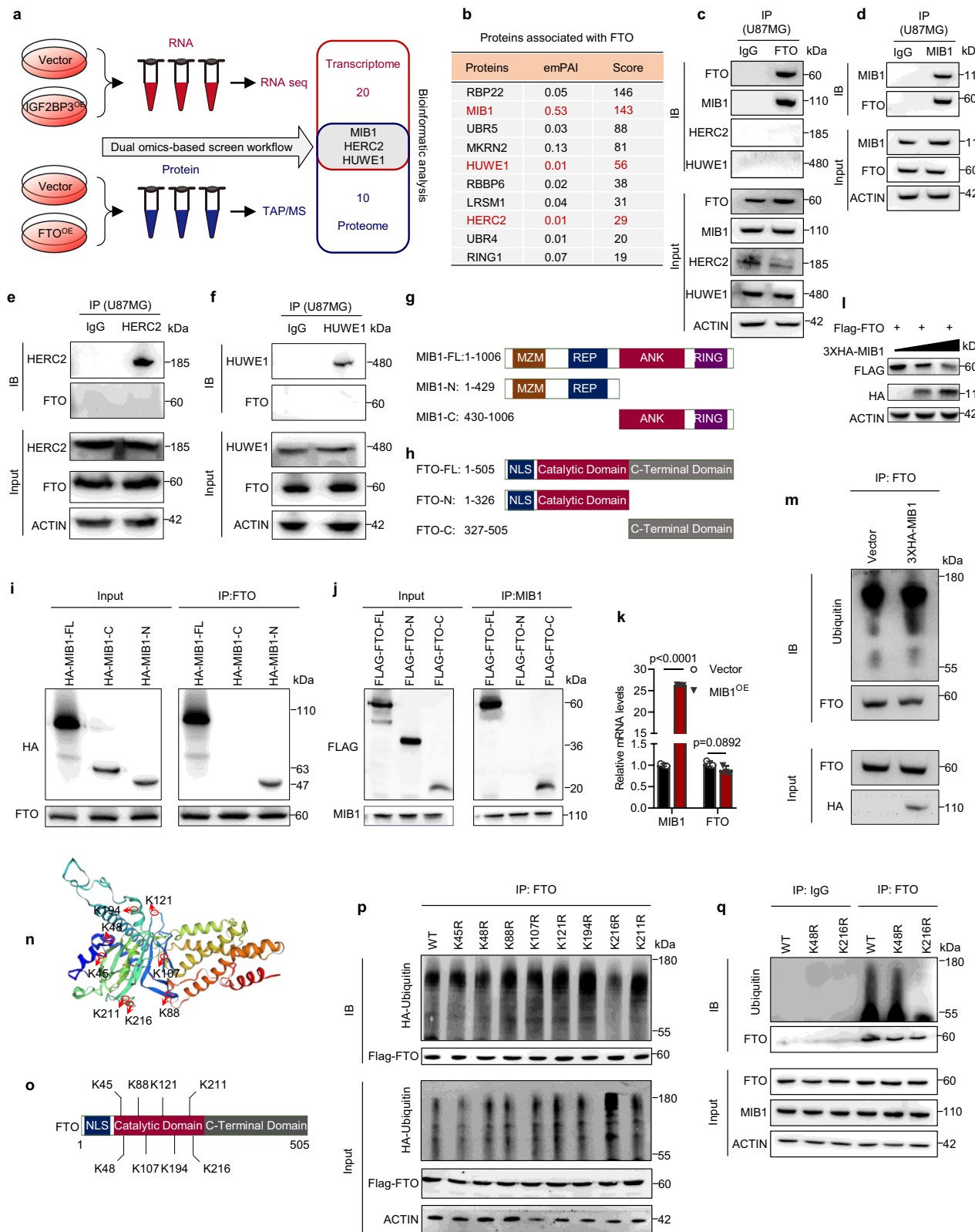

**Nature Communications** | (2024)15:131

## Methods

### Ethics statement

All animal experiments described in this study were approved by the Institute of Animal Care and Use Committee of Fudan University (IACUC no. 20220228-089; 20230301-071). Tumor samples were collected with the patients' written informed consent and approved by the Human Research Ethics Committee of Zhongshan Hospital, Fudan University (no. Y2020-043).

### TCGA data sources

The RNA sequencing data in transcripts per kilobase million (TPM) format for TCGA-LGG (Lower Grade Glioma; WHO grade II–III) and

**Fig. 4 | MIB1 interacts with FTO and mediates FTO protein degradation. a** Dual omics-based screening workflow was employed. For RNA-seq and Mass spectrometry analysis, $n = 3$ biological independent experiments. **b** The ten proteins associated with FTO were identified using mass spectrometry. $n = 3$ biologically independent experiments. **c–f** Immunoprecipitation (IP) and western blot analysis were conducted to validate the interaction between FTO and MIB1, HERC2, and HUWE1 proteins in U87MG cells. $n = 3$ biological independent experiments. **g, h** Schematic diagrams of the MIB1 and FTO fragments used in the immunoprecipitation experiments shown in **i, j. i, j** The IP assay was used to map the interaction domain between MIB1 and FTO proteins. $n = 3$ biological independent experiments. **k** RT-qPCR was utilized to detect the effect of MIB1 transfection on FTO mRNA levels. $n = 3$ biological independent experiments. **l** Western blot analysis

demonstrated the effect of MIB1 transfection on FTO expression. 293 T cells were co-transfected with Flag-tagged FTO and increasing doses of HA-tagged MIB1 for 48 h. $n = 3$ biological independent experiments. **m** The effect of MIB1 transfection on the polyubiquitination modification of FTO protein was analyzed in U87MG cells. $n = 3$ biological independent experiments. **n, o** The molecular structure of the FTO protein was generated using SWISS-MODEL (https://swissmodel.expasy.org/). The predicted ubiquitination sites were labeled using UbPred (http://www.ubred. org/). **p, q** IP and western blot analysis were performed to identify the ubiquitination site of FTO mediated by MIB1. U87MG cells were co-transfected with HA-Ubiquitin and indicated FTO mutants. $n = 3$ biological independent experiments. Statistical significance was determined using two-tailed Student's t-test in **k**. Data represent the mean ± SD. Source data are provided in the Source Data file.

TCGA-GBM (glioblastoma multiforme; WHO grade IV) were uniformly processed by TOIL and downloaded from UCSC Xena (https://xenabrowser.net/datapage/).

## ssGSEA (single-sample Gene Set Enrichment Analysis) score

To investigate the level of infiltration of various immune cell populations, we obtained a set of marker genes defining different types of immune cells from the study by Gabriela Bindea et al.[38]. The ssGSEA algorithm, a rank-based method that calculates a score representing the extent of absolute enrichment of a specific gene set in each sample[39], was employed to measure the relative abundance of each type of immune cell in the glioma TME.

## Cell lines

HEK293T, HL60, HA1800, U87MG, GL261, and T98G cells were obtained from the American type culture collection (ATCC). 05MG and U251MG glioma cells were gifted from Dr. Guangmei Yan (Sun Yat-sen University). They were cultured in DMEM medium containing 1% tri-antibody (Penicillin Streptomycin-Nystatin Solution) and 10% inactivated fetal bovine serum at 37 °C and 5% $CO_2$ in a cell culture incubator. Patient-derived glioblastoma stem cells (GSCs) were gifted from Dr. Jianghong Man (National Center of Biomedical Analysis) and maintained in neurobasal medium supplemented with B-27 supplement, sodium pyruvate, L-glutamine, penicillin-streptomycin, 10 ng/ml basic fibroblast survival factor (bFGF), and 10 ng/ml epidermal survival factor (EGF).

## Patient samples

Glioma tissues and paraffin slices were provided by the Department of Pathology, Zhongshan hospital, Fudan University. The usage of tissues for this study has been approved by the Institutional Review Board of the Zhongshan hospital, Fudan University.

## Construction of recombinant viruses

The construction of recombinant oHSV-1 viruses, which delete ICP34.5 and express a single-chain variable fragment against mouse PD1 (scFv-PD1), was conducted using the BAC system, following the previously described methodology[40]. Briefly, the scFv-PD1 sequence, tagged with His at the C terminus, was inserted between the genes encoding UL3 and UL4 under the control of the CMV promoter. The resulting viruses, with or without scFv-PD1 expression, were designated SR1102 and SR1101, respectively.

## Lentivirus construction and infection

Gene knockout was achieved by utilizing the pSp-CAS9(BB)−2A-Puro(PX459) CRISPR/Cas9 construct (Addgene plasmid #62988). The CRISPR/Cas9 guide RNAs were designed using CHOPCHOP v2[41]. To down-regulate IGF2BP3 endogenously, the shRNA sequences and a negative control shRNA (shNC) were cloned into the pLKO.1-puro vector. The coding sequences (CDS) region of human IGF2BP3 and mouse Igf2bp3 were synthesized and cloned into the pLVX-puro vector for the purpose of overexpressing IGF2BP3.

To produce lentiviruses, 293 T cells were transfected with the lentiviral vector along with packaging plasmids using the HighGene plus Transfection reagent (ABclonal Technology, Wuhan, China). The culture media was collected, pooled, and filtered at 48 and 72 h after transfection. Subsequently, the designated lentivirus was utilized to infect glioma cell lines, and the expression of IGF2BP3 was assessed through real-time quantitative PCR and western blotting. Sequences of oligonucleotides (guide RNA and shRNA) are provided in Supplementary Table 1.

## Reagents and cell transfection

Small interfering RNAs (siRNAs) targeting FTO (siFTO) and scrambled siRNA (siNC) were obtained from RIBOBIO (Guangzhou, China). Glioma cell lines were transfected with plasmids using HighGene plus Transfection reagent (ABclonal Technology, Wuhan, China), following the manufacturer's instructions. For double transfection, two different plasmids (2 μg each) were mixed and co-incubated with 5 μl of High-Gene plus Transfection reagent for 15 minutes at room temperature in 6-well plates. Subsequently, the transfection mix was added dropwise to cells in serum-containing medium. After 6 h of transfection, the transfection medium was replaced with fresh cell survival medium. Sequences of oligonucleotides (siRNA and shRNA) are provided in Supplementary Table 1.

## Animal experiments

Female PAD4[-/-] mice at the age of six to eight weeks were obtained from the Dr. Erwei Song Laboratory. Female C57BL/6 mice, aged six to eight weeks, were purchased from Shanghai Slac Laboratory Animal Co. Ltd. Mice were maintained at an ambient temperature of 20–21 °C, humidity-controlled environment at 60%-70% under a 12-hour light/dark cycle with ad libitum access to water and food. All experimental procedures were approved by the Institute of Animal Care and Use Committee of Fudan University (IACUC no. 20220228-089; 20230301-071). According to the IACUC protocol, the maximal tumor burden is 1500 mm³ permitted by ethics committee. When the tumor volume was up to 1500 mm³, it was considered as the humane endpoint. The mice were randomized at the start of each experiment, and the experiments were not conducted in a blinded manner.

For orthotopic transplantations, $1 \times 10^6$ GL261-luc cells were resuspended in 10 μl of PBS and injected into the right hemisphere of the C57BL/6 mice at specific coordinates. The survival of brain tumors was monitored using bioluminescence imaging. Briefly, the mice were anesthetized with isoflurane, and a solution of 1.875 mg luciferin dissolved in 125 μl of phosphate-buffered saline (PBS; 15 mg/ml) was injected intraperitoneally. After ten minutes, the mice were placed in the IVIS Spectrum CT (PerkinElmer) and imaged for one minute using the camera set to the highest sensitivity. Photons emitted from the brain region were quantified using LivingImage software (Xenogen). Luciferase activity was measured as photons emitted per second.

For subcutaneous transplantations, GL261 cells ($1 \times 10^6$ cells/mouse) were injected into the right flank of C57BL/6 mice. Tumor survival was monitored twice per week in the mice. The tumor size was

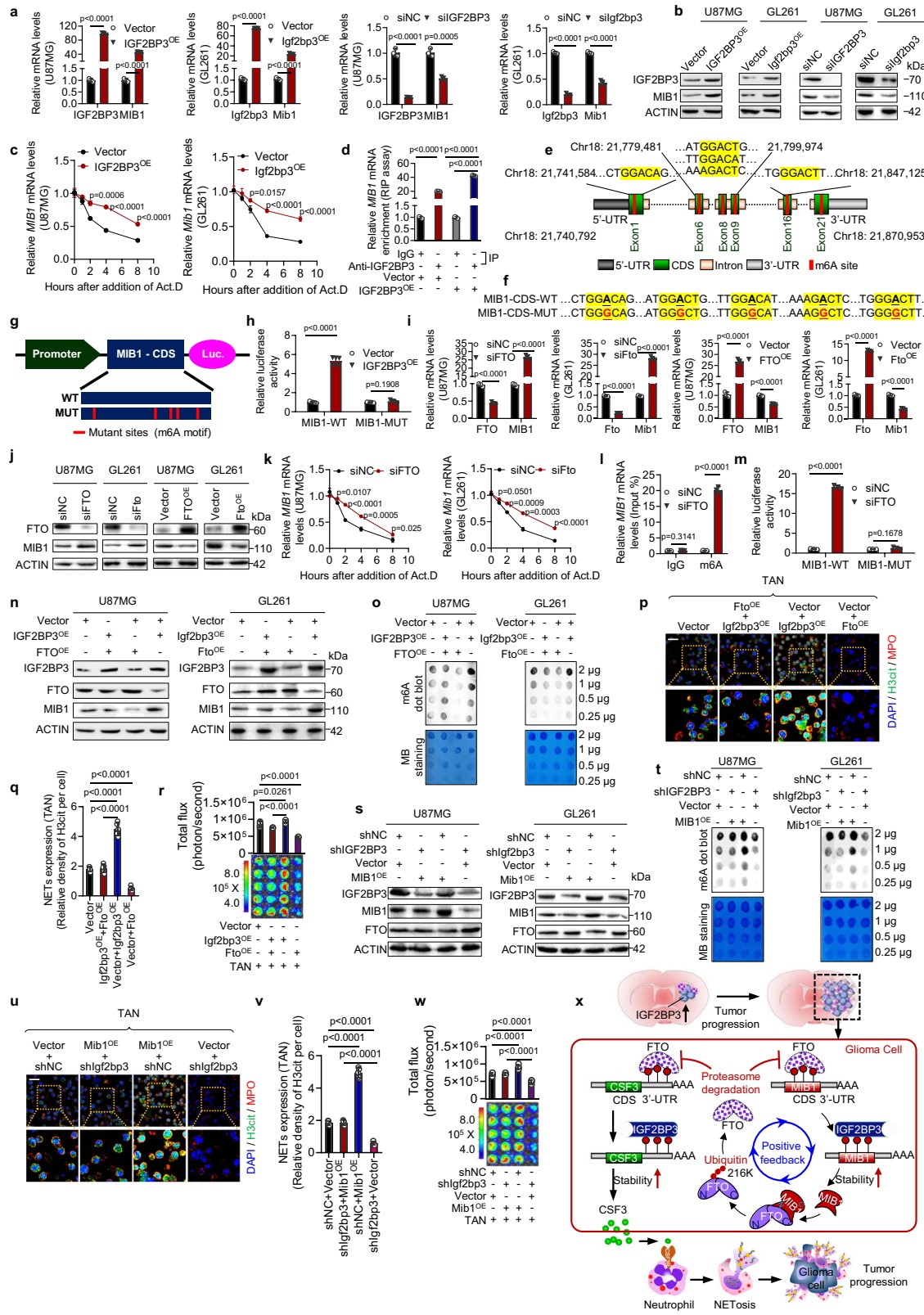

measured in millimeters using a caliper. Tumor volume (V) was calculated as follows: $V = L \times W^2 \times 0.5$; L, length; W, width. At the end-point of the study, the mice were euthanized, and the tumor tissues were weighed.

For the preparation of tumor samples and histology, whole-brain tissue from tumor-bearing animals was fixed with 4% neutral-buffered formalin, embedded in paraffin. The tissue was then cut into 5 μm thick sections and stained with hematoxylin and eosin (H&E).

## Neutrophil isolation (GBM tumors, bone marrow)

Mice were euthanized in accordance with the approved protocol of the institution's animal care committee, using carbon dioxide inhalation followed by cervical dislocation. To isolate tumor-associated neutrophils, intracranial tumors (GBM) were surgically resected and washed three times with sterile-filtered Dulbecco's phosphate-buffered saline (DPBS; 21-030-CVR, Corning). The tumor tissue, ranging between 100 and 200 mg, was minced thoroughly and then

**Fig. 5 | The IGF2BP3/MIB1/FTO pathway creates a positive feedback loop that strengthens NETosis and glioma survival. a**–**c** The impact of IGF2BP3 (Igf2bp3) on the relative mRNA **a** and protein **b** levels of MIB1 and the MIB1 (Mib1) mRNA stability **c**. $n = 3$ biological independent experiments. **d** Evaluation of the interaction between the IGF2BP3 protein and MIB1 mRNA in U87MG cells. $n = 3$ biological independent experiments. **e** The schematic diagram of MIB1 mRNA and the predicted 'm6A' sites with the highest confidence at coding sequence (CDS) are highlighted in red. **f**, **g** The schematic diagram of the luciferase reporter of MIB1. **h**, **m** The relative activity of WT or MUT luciferase reporters, based on the pGL3-Basic plasmid, was assessed in IGF2BP3-transfected U87MG cells **h** and siFTO-transfected U87MG cells **m**, with normalization to control groups ($n = 5$ biological independent experiments for **h** and $n = 3$ biological independent experiments for **m**). **i**, **j** The effect of FTO (Fto) on the relative mRNA **i** and protein **j** levels of MIB1. $n = 3$ biological independent experiments. **k** The effect of FTO (Fto) on the MIB1

(Mib1) mRNA stability. $n = 3$ biological independent experiments. **l** The m6A modification of MIB1 mRNA. The enrichment of m6A was calculated by m6A-IP/input and IgG-IP/input. $n = 3$ biological independent experiments. **n**, **s**, **o**, **t** The effects of FTO interruption **n**, **o** or MIB1 interruption **s**, **t** on the IGF2BP3/MIB1/FTO pathway and IGF2BP3-regulated m6A levels. $n = 3$ biological independent experiments. **p**, **q**, **u**, **v** Representative immunofluorescence images **p**, **u** and quantification **q**, **v** of NET formation in an uncontacted transwell assay of GL261 cells and TANs. Scale bars, 20 μm. Representative images of $n = 5$. $n = 5$ biological independent samples. **r**, **w** The effect of NETs induced by contacted co-culture assay on GL261 cell survival was detected by luciferase assay. $n = 5$ biological independent samples. **x** The proposed working model. Statistical significance was determined using two-tailed Student's t-test in **a**, **c**, **h**, **i**, **k**, **l**, or one-way ANOVA in **d**, **q**, **r**, **v**, **w**. Data represent the mean ± SD. Source data are provided in the Source Data file.

incubated in 15-ml conical tubes containing Hank's Balanced Salt Solution (HBSS; 21-022-CV, Corning) supplemented with 0.1 mg/ml type IV collagenase (C5138, Sigma) and 0.1 mg/ml hyaluronidase (H6254, Sigma) at 37 °C for 15 minutes. The digestion process was halted by adding an equal volume of 2% fetal bovine serum (FBS)/HBSS mixture. The cells were then centrifuged at 1000 rpm (110 × g) for 5 minutes using a ST-40 centrifuge (ThermoFisher) and washed twice with a 2% FBS/HBSS mixture to ensure complete removal of digestive enzymes. Subsequently, the cell suspension was filtered through a sterile 40 μm nylon mesh cell strainer (22-363-547, ThermoFisher) for further purification.

Isolation of neutrophils from the bone marrow of non-tumor-bearing mice was performed based on previously described protocols[42]. For bone marrow extraction, cells were obtained from one femur and one tibia of each mouse by centrifugation at 400 × g for 7 min at 4 °C, following the removal of bone epiphyses. Red blood cells were lysed from the tumor and bone marrow single-cell suspensions using red blood cell lysis buffer (11814389001, Sigma), and the cells were then washed three times with RPMI medium containing 2% FBS. Cell viability was assessed using trypan blue staining and an automated cell counter. Finally, the cells were resuspended in DPBS with 2% FBS before proceeding to immunomagnetic selection or further experimental procedures.

## Luminescence-based cell viability assay
Luciferase-expressing glioma cells (U87/GL261) were seeded at a density of 3000 cells per well in 96-well flat-bottom plates. Cells were either seeded alone or cocultured with neutrophils in a 1:20 ratio. The plates were then incubated at 37 °C for 48 h.

After the coculture period, cell viability was assessed by measuring luminescence using the Luciferase Assay System (E1500, Promega) following the manufacturer's instructions. Briefly, the cells were washed once with DPBS, lysed, and treated with D-firefly luciferin potassium salt dissolved in culture medium. Luminescence was measured using an IVIS baseline reading. Duplicate wells were used for each condition, and the luminescence readings were averaged.

To calculate percent survival, the luminescence values of coculture wells were normalized to that of tumor cell monoculture, after subtracting the background luminescence. For all compound treatment experiments, the tumor cell monoculture treated with the same compound at the same concentration was used as a control for normalization, representing 100% survival.

## CCK8 assay
Cells were seeded onto 96-well plates at a density of 2000 cells per well. Following the indicated treatment, the cells were incubated in 10 μL of Cell Counting Kit-8 (CCK-8) solution diluted in culture media at 37 °C for 1.5 h. After the incubation period, the absorbance of the samples was measured at 450 nm using a microplate reader, in accordance with the manufacturer's protocol.

## Scanning electron microscopy analysis
For in vitro coculture experiments, a sterile 60 × 15 mm Permanox dish (Nalge Nunc International) was seeded with $8 \times 10^5$ tumor cells one day prior to coculture. The coculture was established by adding either mouse neutrophils or human neutrophils (HL-60) in a 1:20 ratio with the tumor cells. The coculture was maintained at 37 °C for 12 h.

As controls, tumor cells in monoculture were included. After the coculture period, the cells were washed twice with warm PBS and fixed with 2.5% glutaraldehyde overnight. The fixed samples were then submitted to the Research Institute of Bioinformatics (Fudan University) for further processing.

In brief, the samples were washed with PBS, incubated with 1% osmium tetroxide, and dehydrated using a graded ethanol series. Subsequently, the samples underwent critical-point drying and were coated with a 2 nm platinum layer. Afterward, a 5 nm carbon coating was applied to the samples. The samples were then analyzed using a Talos L120C scanning electron microscope.

## RNA extraction and quantitative real-time PCR (qRT-PCR) assay
Total RNA was extracted from tissues and cells using Trizol reagent (Invitrogen, USA) following the manufacturer's instructions. Subsequently, cDNA synthesis was performed using the 1st strand cDNA Synthesis kit (Yeason, China). For mRNA quantification, qRT-PCR was carried out using the Hieff® qPCR SYBR® Green Master Mix (Yeason, China). The expression level of β-actin was used as an internal control for mRNA detection. Sequences of oligonucleotides (primers) are provided in the Supplementary Table 1.

## Western blot assay
Total protein was extracted using RIPA buffer (Beyotime, China) supplemented with a protease inhibitor cocktail (Yeason, China). The protein concentration was determined using the bicinchoninic acid (BCA) assay (Yeason, China). The protein samples were separated by 10% SDS-PAGE and transferred onto polyvinylidene fluoride (PVDF) membranes (Millipore, USA). The membranes were then incubated with primary antibodies against METTL3, METTL14, ALKBH5, FTO, MIB1 (all from Proteintech, Wuhan, China) at a dilution of 1:1000. The antibodies against HSV viral proteins ICP0, ICP8, and gC were obtained from Dr. Bernard Roizman's laboratory at The University of Chicago and were used at a dilution of 1:1000. Subsequently, the membranes were incubated with a peroxidase (HRP)-conjugated secondary antibody (1:1000, Cell Signaling Technology, USA). After thorough washing, the protein signals were detected using a chemiluminescence system (Tanon, China) and analyzed using Image Lab Software. The detailed information on the antibodies and related agents used in this study is provided in the Supplementary Table 2.

## Immunoprecipitation (IP) assay
U87MG cells transfected with MIB1 and FTO constructs were cultured for 48 h. Afterward, the cells were harvested and lysed using high-salt

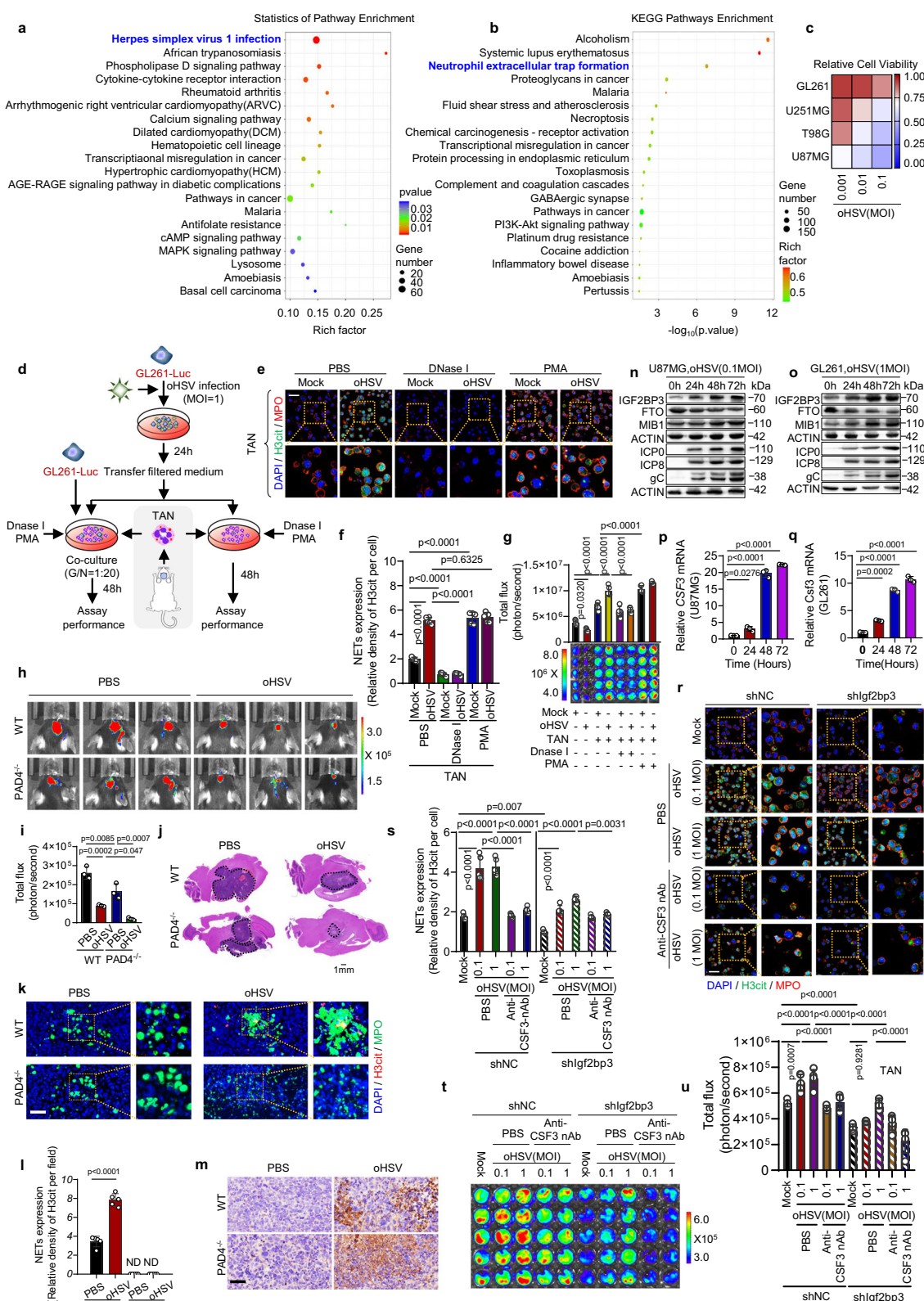

lysis buffer containing 20 mM Tris-HCl (pH 8), 1 mM EDTA, 400 mM NaCl, 1% Nonidet P-40, 0.1 mM sodium orthovanadate, 10 mM NaF, 2 mM DTT, and a protease inhibitor mixture. The lysates were kept on ice for 1 h, and insoluble material was removed by centrifugation. The supernatant was then mixed with an equal volume of lysis buffer without NaCl, resulting in a final NaCl concentration of 200 mM. To reduce non-specific binding, the lysates were precleared with pre-

immune rabbit serum for 2 h at 4 °C. Afterwards, 50 μL of a 50% slurry of protein A agarose conjugate was added to the samples. The mixture was centrifuged, and the supernatant was transferred to new tubes. Mouse monoclonal anti-MIB1 and FTO antibodies (1:1000, ProteinTech, China) were added to the supernatant and incubated overnight at 4 °C to allow immunoprecipitation of the target proteins. The immune complexes were captured using protein A agarose and

**Fig. 6 | oHSV promotes NETosis and tumor survival in an IGF2BP3 dependent manner. a, b** mRNA-seq assay was performed on U87MG cells with or without IGF2BP3 knockout **a** and with or without oHSV infection **b**. The top 20 pathways with the smallest P values were selected for KEGG enrichment analysis in both cases. *n* = 3 biologically independent experiments. **c** Heatmap shows the sensitivity of glioma cell lines to oHSV infection evaluated by cell viability assessed using the CCK8 assay. *n* = 3 biologically independent experiments. **d** Schematic representation of the co-culture system. **e, f** Representative immunofluorescence images **e** and quantification **f** of NET formation in the GL261 cell /TAN co-culture system. Scale bars, 20 μm. Representative images of *n* = 5. *n* = 5 biological independent samples. **g** The NETs-induced GL261 cell survival in the co-culture system was assessed by luciferase assay. *n* = 5 biological independent samples. **h** Representative images of the luciferase signal from C57BL/6 mice. **i** Tumor size was estimated by quantifying luciferase activity (*n* = 3 mice per group). **j** HE staining analysis of tumor volume, with tumor size indicated by the dashed line.

**k, l** Representative immunofluorescence images **k** and quantification **l** of NET formation. Scale bars, 20 μm. "ND"as undetected. Representative images of n = 5. n = 5 biological independent samples. **m** The expression of ICP0 detected by immunohistochemistry. Scale bar, 20 μm. *n* = 3 biologically independent experiments. **n, o** The protein levels of the IGF2BP3/MIB1/FTO pathway and HSV viral proteins were detected. *n* = 3 biological independent experiments. **p, q** The relative mRNA levels of CSF3 (Csf3) were detected. *n* = 3 biologically independent experiments. **r, s** Representative immunofluorescence images **r** and quantification **s** of NET formation in the GL261 cell/TAN co-culture system. Scale bars, 20 μm. Representative images of *n* = 5. *n* = 5 biological independent samples. **t, u** The NETs-induced GL261 cell survival **t** and quantification **u** in the co-culture system were detected by luciferase assay. *n* = 5 biological independent samples. Statistical significance was determined using two-tailed Student's t-test in **l**, or one-way ANOVA in **f, g, i, p, q, s, u**. Data represent the mean ± SD. Source data are provided in the Source Data file.

washed five times with low-salt lysis buffer. The immunoprecipitates were resuspended in 50 μL of disruption buffer, boiled for 5 minutes, separated by electrophoresis on a 12% denaturing polyacrylamide gel, and transferred to a nitrocellulose sheet. The nitrocellulose sheet was then probed with anti-MIB1 and FTO antibodies (both at 1:1000 dilution, ProteinTech, China).

### Quantification of m6A using LC-MS/MS

The quantification of m6A was carried out using LC-MS/MS (Liquid Chromatography with Tandem Mass Spectrometry). In brief, 1 μg of RNA was digested into nucleosides by adding 2.5 U of nuclease P1 (NEB) to a 20 μL buffer containing 10 mM ammonium acetate, pH 5.3, and incubated at 42 °C for 6 h. Following this, 2.5 μL of 0.5 M MES buffer (pH 6.5) and 0.5 U of Shrimp alkaline phosphatase (NEB) were added. The mixture was then incubated at 37 °C for an additional 6 h and subsequently diluted to a final volume of 50 μL. Finally, 5 μL of the resulting solution was injected into LC-MS/MS for analysis. During LC-MS/MS, the mass transitions of m/z 282.1 150.1 (6-mA) and m/z 267.9 136.1 (adenine, A) were monitored and recorded. To establish the calibration curve, pure authentic nucleoside standards (A, 6-mA, Santa Cruz) were run as external standards. The ratios of m6A to A were then calculated based on the obtained data.

### Protein mass spectrometry

The immunoprecipitation samples were stained with Coomassie blue staining solution for 30 minutes, and the explant was eluted overnight. The dye solution was completely eluted until the color on the gel was clean, colorless and transparent. The enriched bands cut on the separation glue after elution were placed in a centrifuge tube and eluted using 3xFlag peptide buffer (150 mM KCl, 10 mM Tris-HCl (pH 8.0), 20% Glycerol, 0.1 mM DTT and 200 μg/ml 3xFlag peptide). The proteins were then reduced using10 mM DTT at 37 °C for 1 h. Subsequently, trypsin was added at a 1:100 (w/w) ratio for digestion overnight at 37 °C. The resulting peptides was separated and analyzed using the nanoElute system (Bruker Daltonics, Billerica, MA) connected to the quadrupole time-of-flight mass spectrometer (TIMS-TOF Pro, Bruker Daltonics, Billerica, MA). The sample was dissolved in 5 μL solvent A (0.1% formic acid in water), and then 1 μL peptide sample was loaded on the chromatographic column (250 mm × 75 μm 1.6 μm C18 resin, IonOpticks). The entire separation process was a 60-min gradient with 2 – 37% solvent B (ACN with 0.1% formic acid) in 0 – 55 min, and was re-equilibrated at 80% B for 5 min. The flow rate was at 300 nL/min, and the column temperature was maintained at 50 °C. For MS analysis, survey full-scan mass spectra were obtained at the range of 100 – 1700 m/z in positive electrospray mode, and with the 100 ms accumulation and ramp time each. The single acquisition cycle (1.16 ms) contained one full TIMS MS scan and 10 parallel accumulation serial fragmentation (PASEF) MS/MS scans. During the PASEF MSMS scanning, the

collision energy increased linearly as a function of mobility from 59 eV at 1/k0 = 1.6 Vs/cm² to 20 eV at 1/k0 = 0.6 Vs/cm². Other mass spectrometer parameters were as follows: intensity threshold was 5000; the range of 1/k0 was 0.7 – 1.3 Vs/cm²; the capillary voltage was 1500 V; the auxiliary gas flow rate was 3 L/min; the ionization temperature was 180 °C. The mass spectra were searched using PEAKS X-Build online (version 1.7) with Human UniProtKB/Swiss-Prot database (Released on December 02, 2021, containing 20 375 protein sequence entries). The precursor mass and fragment error tolerance were set as 15 ppm and 0.05 Da, respectively. The missed cleavages of specific trypsin digestion were controlled up to 2, and the length of the identified peptides was allowed not less than 6 amino acids. The fixed modification of carbamidomethyl on cysteine and variable modifications of oxidation on methionine and deamidation on asparagine were set. The false discovery rates (FDR) of peptide spectrum match (PSM) and protein level was set as less than 1%.

### RNA extraction and library construction

Total RNA was extracted and purified using TRIzol reagent (Invitrogen, Carlsbad, CA, USA) according to the manufacturer's instructions. The quantity and purity of the RNA samples were measured using Nano-Drop ND-1000 (NanoDrop, Wilmington, DE, USA). The integrity of the RNA was assessed using the Bioanalyzer 2100 (Agilent, CA, USA), with a minimum RNA Integrity Number (RIN) of 7.0, and confirmed by electrophoresis on denaturing agarose gel.

Poly(A) RNA was purified from 1 μg of total RNA using Dynabeads Oligo(dT)25-61005 (Thermo Fisher, CA, USA) through two rounds of purification. The purified poly(A) RNA was then fragmented into smaller pieces using the Magnesium RNA Fragmentation Module (NEB, cat.e6150, USA) at 94 °C for 5–7 min. The cleaved RNA fragments were reverse-transcribed into cDNA using SuperScript™ II Reverse Transcriptase (Invitrogen, cat. 1896649, USA). The resulting cDNA was used for the synthesis of U-labeled second-stranded DNAs with E. coli DNA polymerase I (NEB, cat.m0209, USA), RNase H (NEB, cat.m0297, USA), and dUTP Solution (Thermo Fisher, cat.R0133, USA). An A-base was added to the blunt ends of each strand to prepare them for ligation to indexed adapters.

Indexed adapters, each containing a T-base overhang, were ligated to the fragmented DNA. Size selection was performed using AMPureXP beads. After treatment with the heat-labile UDG enzyme (NEB, cat.m0280, USA), the ligated products were amplified by PCR under the following conditions: initial denaturation at 95 °C for 3 min, 8 cycles of denaturation at 98 °C for 15 s, annealing at 60 °C for 15 s, and extension at 72 °C for 30 s, followed by a final extension at 72 °C for 5 min. The average insert size of the final cDNA library was 300 ± 50 bp. Finally, paired-end sequencing (PE150) was performed on an Illumina Novaseq™ 6000 (LC-Bio Technology CO., Ltd., Hangzhou, China) following the recommended protocol from the vendor.

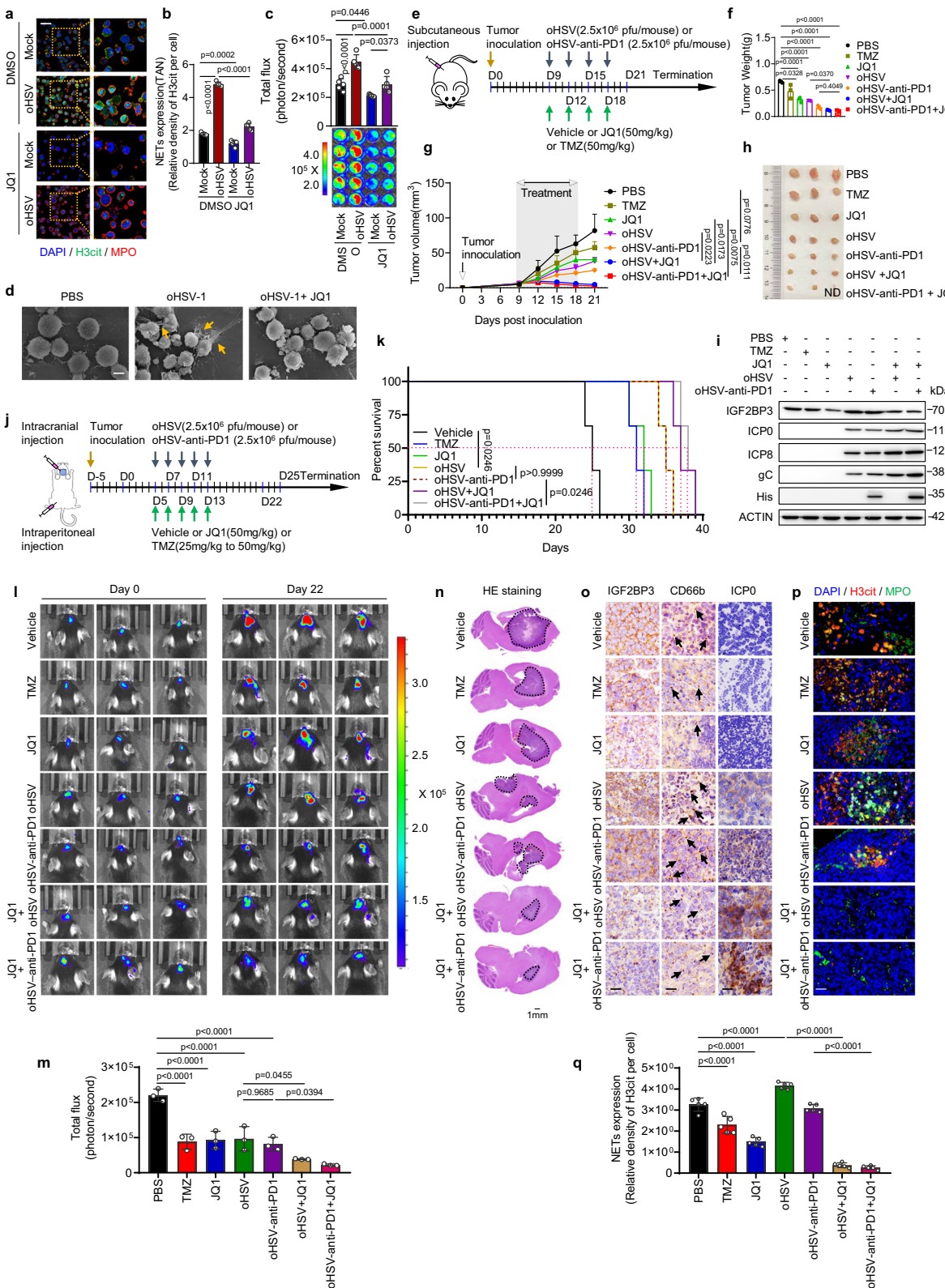

## m6A dot blot assay

Poly(A) RNA was isolated from total RNA using TRIzol reagent (Invitrogen, USA) according to the manufacturer's instructions. The isolated samples (2000 ng, 1000 ng, 500 ng, or 250 ng) were dissolved in SSC buffer and then deposited onto a nitrocellulose (NC) membrane (Millipore, USA). Subsequently, the membrane was crosslinked using UV light for 5 min and stained with 0.02% Methylene blue. The blue dots on the membrane were scanned to visualize the input RNA

content. Next, the membrane was incubated overnight at 4 °C with an m6A antibody (Cell Signaling Technology, USA). After incubation with a secondary antibody, the dot blots were visualized using an imaging system.

## Cycloheximide-based protein stability assay

Cells were treated with 10 µM cycloheximide (CHX) for the indicated periods (0, 2, 4, 8, and 24 h) to block protein synthesis. Additionally,

**Fig. 7 | Pharmacologic inhibition of IGF2BP3 enhances anti-tumor activity of oHSV in glioma. a, b** Representative immunofluorescence images **a** and quantification **b** of NET formation in a co-culture system of GL261 cells and TANs. Scale bars, 20 μm. Representative images of *n* = 5. *n* = 5 biological independent samples. **c** The NETs-induced proliferation of GL261 cells in the co-culture system was detected by luciferase assay. *n* = 5 biological independent samples. **d** Electron microscopic analysis of TANs co-cultured with GL261 cells treated with the indicated agents. Yellow arrows indicate NETs. Scale bars, 5 μm. Representative images of *n* = 3. **e** Schedule of combination treatment for C57BL/6 mice subcutaneously injected with 1 × 106 GL261 cells. **f, g** Tumor weight and tumor survival were measured (*n* = 3 mice per group). **h** Representative images of xenograft tumors, "ND" indicates not detected. **i** Western blot analysis of protein expression levels of IGF2BP3, ICP0, ICP8, and gC. *n* = 3 biologically independent experiments. **j** Schedule of combination treatment for C57BL/6 mice intracranially injected with

1 × 10[6] GL261 cells. **k** Kaplan-Meier analysis was performed on mouse models with intracranially implanted tumors derived from GL261 and treated with the indicated agents to assess survival (*n* = 3 mice per group). **l** Representative images of the luciferase signal from C57BL/6 mice inoculated with GL261 tumors and treated with the indicated agents. **m** Tumor size was estimated by quantifying luciferase activity in tumor cells (*n* = 3 mice per group). **n** HE staining analysis of tumor volume. The tumor size was indicated by a dashed line. **o** Immunohistochemistry analysis of IGF2BP3, ICP0, and neutrophil infiltration (CD66b, labeled with black arrows) in the indicated tumor tissues. Scale bars, 20 μm. **p, q** Representative immunofluorescence images **p** and quantification **q** of NET formation in the indicated tumor tissues. Scale bars, 20 μm. Representative images of *n* = 5. *n* = 5 biological independent samples. Statistical significance was determined using one-way ANOVA in **b, c, f, m, q**, or two-way ANOVA in **g**, or the log-rank (Mantel-Cox) test in **k**. Data represent the mean ± SD. Source data are provided in the Source Data file.

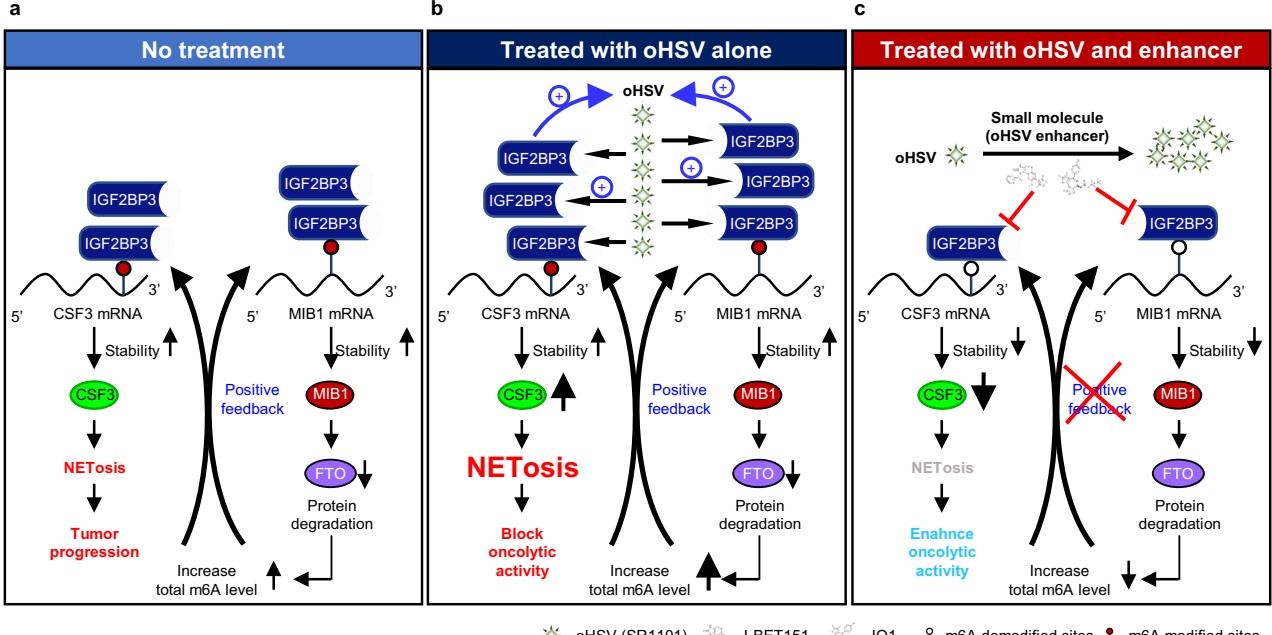

**Fig. 8 | Modeling the role of IGF2BP3-induced NETosis in oncolytic virotherapy of glioma cells. a** IGF2BP3 is upregulated in glioma cells, which leads to an overall increase in m6A levels through the IGF2BP3/MIB1/FTO positive feedback loop, ultimately resulting in CSF3-mediated NETosis and tumor progression. **b** oHSV treatment increases its replication by upregulating IGF2BP3, which triggers immune suppression by promoting NET formation and blocks the oncolytic activity. **(c)** Blocking IGF2BP3 pharmacologically with small molecule (oHSV enhancer) enhances oHSV's oncolytic activity by impeding NET formation.

20 μM MG132 or chloroquine was administered to inhibit protein degradation before harvesting. Crude extracts were prepared, and the protein expression of IGF2BP3 and FTO was assayed following previously described methods.

### mRNA stability detection
At 48 h after transfection, U87MG and GL261 cells were co-incubated with actinomycin D (ActD, 5 μg/ml, Sigma-Aldrich). After ActD treatment with the indicated times, total RNA was extracted from U87MG and GL261 cells, and the mRNA levels of MIB1 and CSF3 were measured using an RT-qPCR assay.

### Dual-luciferase reporter assay
Cells were co-transfected with plasmids containing wildtype or mutant fragments from MIB1 CDS and CSF3 3′-UTR, along with pcDNA3.1-IGF2BP3 or siFTO, using HighGene Plus Transfection reagent (ABclonal Technology, Wuhan, China) following the manufacturer's protocol. At 24 h after transfection, firefly and renilla luciferase activities were measured consecutively using the dual-luciferase reporter assay

system (Promega, USA). Finally, the ratios of luminescence from firefly to renilla luciferase were calculated.

### RNA immunoprecipitation (RIP)
10 million cells were harvested and washed with cold PBS, followed by UV crosslinking at 254 nm. Cells were then lysed using 1 mL of RIP buffer (150 mM KCl, 25 mM Tris, pH 7.4, 5 mM EDTA, 0.5 mM DTT, 0.5% IGEPAL, 100 U/mL RNAase inhibitor SUPERasedin, and a cocktail of proteinase inhibitors), and sonicated for 10 cycles with 30 s on and 30 s off for each cycle using a Bioruptor (Diagenode). Ten percent of the lysate was saved as the input sample. The remaining lysate was subjected to preclearing with protein A/G magnetic beads and then incubated with anti-IGF2BP3 antibody at 4 °C for 2 h. Subsequently, protein A/G magnetic beads were added and the mixture was incubated overnight at 4 °C with rotation. The beads were washed three times with RIP buffer. The immunoprecipitated samples were subjected to DNase and Proteinase K digestion, followed by total RNA isolation using the TRIzol reagent in parallel with the input samples.

## m6A RNA immunoprecipitation assay

MeRIP (methylated RNA immunoprecipitation) was conducted using the Magna MeRIP m6A Kit (Millipore, USA) following the manufacturer's instructions. In brief, 3 μg of anti-m6A antibody (Abcam, USA) was conjugated to protein A/G magnetic beads overnight at 4 °C. The antibody-conjugated beads were then incubated with the antibody in IP buffer containing RNase inhibitor and protease inhibitor. The RNAs that interacted with the antibody were isolated and detected using RT-qPCR.

## Immunohistochemistry assay

The tissues were fixed overnight at 4 °C in 4% paraformaldehyde in PBS and subsequently embedded in paraffin wax. Paraffin-embedded tissues were cut into 5 μm sections, and after deparaffinization and rehydration, heat-induced antigen retrieval was performed using 0.01 M citrate buffer (pH 6.0). Endogenous peroxidase activity was blocked by treating the sections with 3% hydrogen peroxide for 15 min at room temperature in the dark. To prevent nonspecific binding, the sections were blocked with 5% serum for 1 h at room temperature. Primary antibodies(rabbit anti-IGF2BP3, ABclonal, Cat.: #A4444, dilution 1:200; rabbit anti-CD66b, Proteintech Group, Cat.: #19496-1-AP, dilution 1:200; mouse anti-ICP0, provided by Dr. Bernard Roizman (University of Chicago), dilution 1:200) diluted in blocking solution were applied to the sections and incubated overnight at 4 °C. Afterward, the sections were incubated at room temperature for 1 h with secondary antibodies, followed by the addition of an avidin-biotin peroxidase complex. DAB reagent was used for staining, and counterstaining was performed with hematoxylin. Finally, images were captured using an upright microscope system (Nikon, JAPAN).

## Activation of neutrophils and NET formation assay

Isolated neutrophils were cultured in 24-well plates with 500 μL of serum-free DMEM. To induce the formation of neutrophil extracellular traps (NETs), the neutrophils were activated with PMA for 6 h. DNase I was added 6 h prior to neutrophil activation to digest NET DNA. To induce NETs in the presence of cancer cells, a coculture system was established using transwell chambers with a 0.4 μm porous membrane (#353495, Corning) separating the two chambers. Neutrophils were placed in the bottom chamber, while the cancer cells (GL261) were placed on the membrane of the upper chamber. After 48 h, the neutrophils were collected.

To assess NET formation, neutrophils grown on poly-l-lysine-coated coverslips (#354085, Corning) were fixed with 4% paraformaldehyde (PFA) for 20 min at room temperature. The cells were then rinsed three times in PBS. Next, the cells were blocked in PBS containing 1% bovine serum albumin (BSA, #A3294, Sigma) for 30 min and incubated overnight at 4 °C with rabbit anti-H3cit antibody (1:100, Abcam, ab5103) and mouse anti-MPO antibody (1:100, ProteinTech, 666177-1-Ig) in blocking buffer. After three washes in PBS, the cells were incubated with fluorochrome-conjugated secondary antibodies (1:200, Cell signaling technology) for 1 h, rinsed twice in PBS, stained with 4′,6-diamidino-2-phenylindole (DAPI, #D1306, Thermo Fisher Scientific) for 5 min, rinsed in water, and the coverslips were mounted onto glass slides using mounting media. The images were captured using laser scanning confocal microscopy (F1300032, Leica).

## Immunofluorescence staining

The tissue was fixed in 4% paraformaldehyde (Thermo Scientific) at 4 °C for 24 h, followed by washing with PBS. After embedding in paraffin, the tissue was sectioned at a thickness of 4 μm. Antigen retrieval was performed using target retrieval solution, pH 9.0 (Dako), in a pressure cooker for 15–20 min. To block non-specific binding, 5% BSA was applied for 25 min at room temperature. For immunofluorescence of cells, they were fixed with 4% paraformaldehyde for 25 min at room temperature, washed with PBS. Subsequently, the cells were blocked in PBS containing 2% BSA for 30 minutes at room temperature. The samples were then incubated overnight at 4 °C with rabbit anti-H3cit antibody (1:100, Abcam, ab5103) and mouse anti-MPO antibody (1:100, ProteinTech, 666177-1-Ig). For tissue samples, they were incubated with Alexa-Fluor-conjugated secondary antibodies (Invitrogen) in 1% BSA for 1 h at room temperature. DAPI was used for nuclear counterstaining, and images were captured using laser scanning confocal microscopy (LSM800, Zeiss). The percentage of positive H3cit signal in each field of view in the overall tissues was determined as the NETs, analyzed using Imaris 9.0 Microscopy Image Analysis Software. NETs were quantified by counting in at least 10 fields per section, and 5 sections per sample were evaluated.

## Chromatin immunoprecipitation (ChIP) assay

A modified ChIP assay was performed to investigate protein and viral DNA interactions. Firstly, HSV-1-infected cells ($2 \times 10^7$) were washed with ice-cold PBS three times and then cross-linked using 1% formaldehyde. The cross-linking reaction was quenched by adding glycine (125 mM). After rinsing with ice-cold PBS, the cells were collected and lysed in a cell lysis buffer containing 5 mM PIPES (pH 8.0), 1% SDS, 1 mM EDTA, and protease inhibitors. The mixture was incubated on ice for 15 min, followed by sonication to shear the DNA using a 15-s on and 10-s off cycle at an amplitude of 30 for a total of 20 min. The supernatants containing chromatin were collected by centrifugation and used immediately or stored at −80 °C. For immunoprecipitation, the supernatants were diluted with ChIP dilution buffer (Beyotime) and pre-cleared using salmon sperm DNA/protein G agarose beads (Roche). The protein-DNA complexes were then immunoprecipitated using specific antibodies and protein G beads at 4 °C overnight. The immune complexes were washed sequentially with a low salt buffer (containing 150 mM NaCl in buffer A: 2 mM EDTA, 1% TritonX-100, 0.1% SDS, 20 mM Tris-HCl, pH 8.1), a high salt buffer (containing 500 mM NaCl in buffer A), and a LiCl wash buffer (containing 250 mM LiCl, 1 mM EDTA, 1% NP-40, 1% deoxycholate, 10 mM Tris-HCl, pH 8.1). The protein-DNA complexes were eluted using an elution buffer (containing 1% SDS, 0.1 M NaHCO3). Contaminating RNA was removed by treating with 10 μg/ml RNase A. The cross-links were reversed by incubating the complexes at 65 °C overnight, followed by proteinase K treatment at 55 °C for 1 h to remove proteins. The DNA was purified using phenol chloroform extraction, ethanol precipitation, and then used for DNA-protein association analysis using real-time PCR. Quantitative PCR (qPCR) was performed using the following primers:

ICP0: ATAAGTTAGCCCTGGCCCCGA and GCTGCGTCTCGCTCCG
UL29: CCACGCCCACCGGCTGATGAC and TGCTTACGGTCAGGTGCTCCG
UL41: ATACCATAATTTTATTGGTGGGTCG and CGACCCACCAATAAAATTATGGTAT
GAPDH: TTCGACAGTCAGCCGCATCTTCTT and CAGGCGCCCAATACGACCAAATC
IFN-β: TAGTCATTCACTGAAACTTTA and AGGTTGCAGTTAGAATGTC

## Statistics and reproducibility

All grouped data were presented as mean ± standard deviation (SD), unless otherwise specified. All in vitro experiments were repeated at least three times. Differences in means between groups were analyzed using the two-tailed unpaired Student's t-test for two-group parametric data or the Mann-Whitney U test for two-group non-parametric data. For multiple-group data, one-way analysis of variance (ANOVA) was employed. Statistical significance was defined as a probability value of 0.05 or less. Survival analysis was conducted using Kaplan-Meier survival curves, and differences between different patient or mouse groups were assessed using log-rank statistics. GraphPad Prism 9 Software (GraphPad Software, Inc.) was used for all statistical analyses.

**Reporting summary**

Further information on research design is available in the Nature Portfolio Reporting Summary linked to this article.

## Data availability

The sequencing data generated in this study have been deposited in the GEO repository under accession codes GSE235828 and GSE235568.

The raw data from the protein mass spectrometry experiments generated in this study have been deposited in the ProteomeXchange under accession code PXD043362. The IGF2BP3 publicly available transcriptome data used in this study are available in the TCGA database (https://portal.gdc.cancer.gov. Assessed on 19 July 2019). The authors declare that the remaining data generated or analyzed during this study are available within the article, Supplementary Information, or Source Data file. Source data are provided in this paper.

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

## Acknowledgements

We thank Dr. Erwei Song (Sun Yat-sen University) and Dr. Huiyan Li (Nanhu Laboratory, National Center of Biomedical Analysis) for the generous gifts of PAD4$^{-/-}$ mice and plasmids encoding ubiquitin mutants, respectively. We thank Dr. XueMin Zhang (Nanhu Laboratory, National Center of Biomedical Analysis) for helpful discussions. This work was supported by the National Natural Science Foundation of China (No. 82073880 and 81872467), Shanghai Municipal Health Commission (No. 201940233), Program for Professor of Special Appointment (Eastern Scholar) at Shanghai Institutions of Higher Learning (No. TP2019047).

## Author contributions

M.S. (lead contact) conceptualized the project. W.D. performed most of the experiments with the assistance of Y.C., L.Y., Y. L., S.C., B.Y., R.Y., and Z.F. W.D. and L.Y. performed mouse studies. W.D., Y.C., S.B., and B.Y. performed sequencing data analysis. Y.H. provided the patient samples. W.D., S.B., and R.T. contributed to interpreting the data. X.P. contributed to proofreading the manuscript. R.L., J.C., contributed critical reagents. M.S. (lead contact) designed and supervised the study, analyzed the data, and wrote the manuscript.

## Competing interests

No potential conflict of interest to disclose.
