## [Peer Review File · Nature Communications]

Overcoming Therapeutic Resistance in Oncolytic Herpes Virotherapy by Targeting IGF2BP3-Induced NETosis in malignant gliomaREVIEWER COMMENTS

Reviewer #1 (Remarks to the Author): with expertise in oncolytic viruses, brain tumors

This is a very exciting and novel study that shows that IGF2BP3 expression correlates with netosis. They further show that netosis formation requires CSF3 production and is abrogated by anti CSF3 antibodies. The investigators have further identified that m6A induction by IGF2BP3 is mediated by reduction in MTO via induction of Mib 1 activity and expression.

Overall the data is rigorous and uses multiple different techniques to show netosis: IF, EM etc and the findings are novel and translationally relevant.

Some suggestions to further enhance the study are below.

1. Figure 7E shows gl261 cells injected in BalbC. Gl261 cells are syngeneic to C57Bl mice not BalbC mice. This should be corrected if this was an error, or the data removed.
2. The mechanism of how HSV infection induces netosis is also a novel finding to my knowledge. Apart from IGF2BP3 induction, HSV treatment also induces MIB 1 activity, to induce NOTCH signaling. This should be discussed.
3. It will be interesting to see if the mutant FTO that can not be ubiquitinated by MIB1 can rescue the effect on CSF3 induction and m6A production.
4. Very minor: Fig 1D is called out in the text after fig 1E.

Reviewer #2 (Remarks to the Author): with expertise in neutrophils/NETs

In this manuscript, the authors use in vitro and in vivo experiments to investigate the mechanism of resistance against oHSV. The first part of the manuscript focuses on the role of IGF2BP3 and NETs in the growth of glioma cells. The second part focuses on the identification of the IGF2BP3/MIB1/FTO axis and positive feedback loop. The last part of the manuscript focuses on the role of the IGF2BP3/NETs pathway in resistance to oHSV.

Overall strength:

The second part of the study (the identification of the IGF2BP3/MIB1/FTO axis and loop) is convincing. The role of NETs in therapy response is getting more and more attention, however, to my knowledge, this is the first time that NETs are involved in resistance to

oHSV.

Overall weaknesses:

Control neutrophils (without cancer cells) are never used in vitro, making it impossible to interpret the results generated by the authors. The fact that IGF2BP3 overexpression is increasing the number of cancer cells (without neutrophils) using in vitro assays is shown; however, it is never discussed by the authors. Nevertheless, this is a key result and it is important to take this result into account to interpret correctly the data generated on the role of IGF2BP3-induced NET formation and the role of this axis on cancer cell growth. Indeed, the data shown suggest a role of IGF2BP3 on cancer cell growth independently of NET formation.

As IGF2BP3 is inducing cancer cell growth (in a NET-dependent, and non-NET dependent manner), it is difficult to interpret the results on the role of IGF2BP3 on oHSV resistance. Figures and panels are not listed in order (and sometimes figures are not referenced correctly in the manuscript), making it very hard for the reader to follow and understand the experiments easily. In addition, the organization of the panels within a figure makes it difficult to read the figures correctly.

The work from the authors is of interest and represent another step toward the understanding of NETs in resistance to therapy. However, the main issue I have for publication in Nature Communications is 1) the lack of clarity between the text and the figures and; 2) the lack of proper interpretation of the results.

Specific concerns:

Major concerns:

1) Fig. S1H is cited before S1D. It is the same thing for Fig. S1K and S1L which are cited before S1F. This makes it difficult to read. It is the same thing for a lot of main and

supplemental figures which are not cited in order. The organization of the different panels makes it difficult to read the data easily (see Fig. 5S, T, U, V, W, X; these panels do not follow a logical organization within the figure).

2) In Figs. 1B, C, 1E, S1D, S1E, S1J, S1K and S1L, the authors are not including control neutrophils (unstimulated, without cancer cells), therefore it is impossible to conclude anything from the experiments. This is especially problematic for Fig. 1B/C, in which the authors are using TANs.

3) In Figs. 1D, S1F and S1I, IGF2BP3 overexpression alone increases the number of cells at the end of the experiment, which is therefore independent on neutrophils. Therefore, the author cannot conclude that IGF2BP3-induced NETosis promotes glioma cell growth in an IGF2BP3-dependant manner. This result that IGF2BP3 alone is inducing an increase in the number of cells is key to interpret all the results from these experiments, however the authors do not consider this key result.

4) As IGF2BP3 is promoting an increase in the number of cells in vitro, it is impossible to conclude that the effect of IGF2BP3 modulation in vivo (overexpression vs downregulation) is due to the effect of NETs. Indeed, in PAD4 KO mice, overexpression of IGF2BP3 in cancer cells do promote an increase in tumor size, indicating that in vivo, the effect of IGF2BP3 is also partly independent on NETs. This needs to be discussed clearly in the results. Also, the authors need to show at the mRNA or protein level the overexpression/downregulation of IGF2BP3 in the cancer cells used in the study (for example the cells used in Fig. 1F).

5) The authors conclude an effect on the growth of the cancer cells, but growth specifically is not assessed (indeed the authors only show a bioluminescence signal, which should correlate with the number of cells in the assay). Indeed, the word "Tumor growth" is essentially used when concluding on the in vitro/in vivo experiments. Cells need to proliferate to grow, however proliferation of the cancer cells is never shown. A marker of proliferation on key experiments should be used if the authors want to conclude an effect on the "growth" of the cancer cells.

6) In the in vitro assays presented, the authors need to compare the basal level of NET formation between TANs and control neutrophils (from the bone marrow and/or the non-tumor tissue). The authors also need to compare the effect of TAN and control neutrophils on cancer cell growth in the same experiment.

7) Figs. 1F and 1G have not been performed at the same time, therefore it is impossible to conclude anything on the role of PAD4.

8) In Figs. 2Q-T, the authors show that overexpression of IGF2BP3 in cancer cells alone is sufficient to increase the number of cancer cells in the assay, suggesting a role independent of NETs. Again, this needs to be discussed. The experiments from Figs. 2S and ST need to be performed at the same time as they are comparing WT vs PAD4 KO mice, these should not be independent experiments.

9) In Fig. 6G, it would be important to assess the effect of oHSV alone, without neutrophils.

10) As IGF2BP3 is inducing cancer cell growth (in a NET-dependent, and non-NET dependent manner), it is difficult to interpret the results on the role of IGF2BP3 on oHSV resistance. Indeed, proliferation and resistance are probably taking place at the same time.

11) oHSV efficacy is in part mediated by antitumor immunity. From the experiments presented here, the authors never look at this phenomenon. Therefore, I think it is important to mention that the effect of NETs on oHSV resistance is independent on the regulation of antitumor immunity (the opposite is suggested in the discussion). For example, the authors wrote: "our findings revealed that oHSV treatment induces an OVIS effect". However, the authors never show that oHSV promotes immunosuppression. The same comment applies for the abstract, in which OVIS is mentioned.

If the authors want to show immunosuppression, they must properly show it using their in vitro and in vivo model, adding dendritic cells and CD8 T cells to the system for example. However, I don't think this is needed, the authors just need to tone down these interpretations of the results.

Minor concerns:

12) In the introduction, the authors mention that the implications of NETosis for tumor immunotherapies are not yet fully understood, however a few papers describe a role for NETs in immunotherapy response and these papers need to be cited (for example Zhang et al, J Exp Med, 2020 and Teijeira et al, Immunity, 2020).

13) It is totally unclear from the text and the legends when a proper co-culture assay or a transwell assay (as depicted in Fig. 1A for example) is used. For example, in Fig. 1B, it is written in the legend that a co-culture system is used, however all the cells are positive for MPO, suggesting that no cancer cells are present in this co-culture.

14) In Fig. 1M, the number of neutrophils also needs to be quantified. Indeed, in PAD4 KO mice there is a decreased number of neutrophils compared to the WT mice. This is in accordance with a lot of published papers indicated that when NET formation is inhibited, the recruitment of neutrophils is somehow limited, suggesting a role of NETs for neutrophil recruitment at the inflammatory site.

15) In Fig. 2J-P, control neutrophils without cancer cells need to be added to the experiment. I think it is important to mention that CSF3 is also named G-CSF, which have been shown before to induce NET formation in cancer (Park et al, Science Translational Medicine, 2016).

16) In Fig. 1T, NETs are not quantified.

17) Can oHSV induces NETs ? Authors refer to Fig. S3N-P, however Fig. S3 stops at panel M. This kind of error makes the paper really difficult to read.

18) Some key experiments could be repeated with a co-culture between U87MG sensitive cells and neutrophils.

19) Can stimulation of cancer cells with purified NETs have an effect on the IGF2BP3/MIB1/FTO pathway?

20) Fig. 7P, JQ1 + oHSV treatment has a strong effect on the presence of neutrophils, this needs to be discussed.

21) In the discussion the authors discuss the role of NETs in cancer progression, but they should focus on the role of NETs in resistance to treatment, as the focus of their paper is on resistance against oHSV therapy. Indeed, a few papers now show an important role for NETs in immunotherapy and chemotherapy treatment.

Reviewer #3 (Remarks to the Author): with expertise in brain tumors, cancer immunology

In the first part of the current manuscript, authors demonstrate that IGF2BP3 expression is increased in gliomas, and it increases the expression of MIB1, leading to degradation of FTO via ubiquitin-proteasome pathway, resulting in the increased m6A-mediated release of CSF3 and the formation of NET. NETs promote glioma progression. In the second part, the authors show that treatment with oHSV increases the expression of IGF2BP3, which leads to increased Nets and immunosuppression limiting oHSV treatment efficacy. Further, authors show that BET inhibitor enhances oHSV oncolytic activity by inhibiting IGF2BP3 induced NETs while at the same time reinforcing virus replication via inducing BRD4 recruitment to CDK9/RBP-1 complex to HSV gene promoter. Overall, this is a tour de force effort- presented data are sufficient for two manuscripts.

The first part is exciting, and there is a lot of high-quality data; the innovation is high since NETs and neutrophils are generally understudied in GBM. It is a nice and logically dissected mechanistic study. It stands as a manuscript on its own.

All the experiments use U87 or GL261 (highly immunogenic); both are suboptimal for immune-related studies. Authors have primary GBM cultures and should consider repeating some critical findings with primary GBM patient-derived lines.

There is limited data with the melanoma model, and it does not add much. Authors should

consider removing it, or if they want to keep critical experiments, they should also be performed in the melanoma model.

Minor comments-

1. For BLI imaging subfigures Y-axis is labeled ROI- should be photon/per second. Some of the scale bars are impossible to see.
2. All the Kaplan-Meier survival experiments should include the number of animals and median survival times.
3. Culturing neutrophils, especially from tumors, is a major challenge for the field. Therefore, more details should be provided on experimental procedures for all neutrophil culture experiments. Ideally, detailed protocols should be attached as supplementary files.

The second part with oHSV is not impressive. Results in Figure 7 show the effect on animal survival is marginal. Statements should be toned down to reflect what the data show, including the title of the manuscript.

1. In addition, it is unclear why particular treatment time points were selected – for intracranial five days post-injection and s.q. tumors nine days post-injection?
2. S.Q tumors are treated when the tumor is undetectable (Fig. 7G); usually, these should be treated at a minimum size of 100mm³.
3. Figure 7E BALB/C mice are incorrect for illustration; it should be BL6 for GL261.
4. For Intracranial tumors, MRI should be performed to ensure sizeable tumors in mice and that equal size tumors are distributed in various groups.

Reviewer #4 (Remarks to the Author): with expertise in m6A, brain tumors

This study determines that the positive feedback loop of IGF2BP3/MIB1/FTO can enhance NETosis which reduces the oncolytic activity of oHSV, promoting glioma progression. Furthermore, the authors also demonstrate that pharmacologic inhibition of IGF2BP3 blocks NETosis and enhances anti-tumor activity of oHSV in glioma which has certain clinical translational potential. However, there are some issues need to be addressed to make this work more rounded.

Major points

- 1、 In this study, authors established three in vitro co-culture models of gliomas cells with neutrophils of mice. However, the co-culture models using tumor cells and neutrophils from glioma patients may make the study more convincing. Please conduct co-culture models with tumor cells and neutrophils from patients to further verify the conclusion.
- 2、 In this study, authors demonstrated that U87MG were the most sensitive to oHSV. This phenomenon may depend on the specificity of U87MG. Please add another glioma cell line in the previous experiment to further confirm that the feedback loop of IGF2BP3/MIB1/FTO can enhance NETosis.
- 3、 In Fig 3c, the change of METTL3 expression was also significant after IGF2BP3 knocking out or overexpressing. However, in m6A modification, METTL3 play the opposite role to FTO. Please design experiments to confirm that METTL3 does not play a major role in IGF2BP3-mediated NETosis.
- 4、 In Fig3, authors confirmed that IGF2BP3 can not affect the mRNA level of FTO. Please confirmed that IGF2BP3 can not bind to the mRNA of FTO。
- 5、 In Fig3, please design rescue experiment to demonstrate that IGF2BP3 regulates the expression of CSF3 through FTO.
- 6、 In Fig5, authors demonstrated that FTO can regulate the m6A modification of MIB3.IGF2BP3 also has m6A sites. Please verify whether FTO can regulate the m6A modification of IGF2BP3 and affect its expression.
- 7、 Please explain the reason in discussion that U87MG were the most sensitive to oHSV in four glioma cell lines.

Minor points

- 1、 There are too many results in each page of Fig. Please adjust the Fig structure and put some results into the Supplement Fig to increase the reading experience of readers.
- 2、 In Fig 4D、 E、 F et al, please indicate the cell line used for the Western-blot assay.
- 3、 Pease add statistical analysis of western-blot assays.

Dear Reviewer,

Thank you very much for your kind comments and constructive suggestions on our manuscript titled 'Overcoming Therapeutic Resistance in Oncolytic Herpes Virotherapy by Targeting IGF2BP3-Induced NETosis in Malignant Glioma' (Manuscript ID: NCOMMS-23-27782). These comments are all valuable and extremely helpful for refining the manuscript. We have meticulously revised the manuscript based on your recommendations, as well as those of other reviewers, and the revisions have been shown in the manuscript text file with track changes. We hope that the revised manuscript now meets all the necessary requirements.

Regarding the specific advice you provided earlier, our response is as follows.

Reviewer #1 (Remarks to the Author): with expertise in oncolytic viruses, brain tumors

Comment 1: Figure 7E shows gl261 cells injected in BalbC. Gl261 cells are syngeneic to C57Bl mice not BalbC mice. This should be corrected if this was an error, or the data removed.

Response 1: Thank you for your careful review. We have corrected the mislabeled illustration in our revised manuscript. Please review it.

Comment 2: The mechanism of how HSV infection induces netosis is also a novel finding to my knowledge. Apart from IGF2BP3 induction, HSV treatment also induces MIB 1 activity, to induce NOTCH signaling. This should be discussed.

Response 2: Thank you very much for your recognition of our work and your valuable suggestion. Indeed, MIB1 plays a critical role in activating the NOTCH signaling pathway by ubiquitinating and endocytosing NOTCH ligands. Notch signaling is well-known for its involvement in numerous conserved signal transduction pathways related to immune cell development and homeostasis. It has also been reported that monocytes, macrophages, and dendritic cells (DCs) constitutively produce NOTCH ligands and receptors, enabling them to initiate and respond to NOTCH signals through Toll-like receptor (TLR) regulation. Therefore, it is possible that the HSV-induced upregulation of MIB1 may also lead to the activation of the NOTCH signaling pathway, contributing to TME regulation. We have incorporated the potential connections between MIB1/NOTCH and TME regulation into the discussion section of our revised manuscript. Please review it.

Comment 3: It will be interesting to see if the mutant FTO that cannot be ubiquitinated by MIB1 can rescue the effect on CSF3 induction and m6A production.

Response3: Thank you for your professional advice. We have tested the effect of FTO-216R mutant that cannot be ubiquitinated by MIB1 on the CSF3 induction and m6A production, and we found that FTO-216R mutant dramatically blocked the IGF2BP3-induced upregulation of CSF3 and the total m6A level (see Supplementary Fig. 2M-R). We have added the data in our revised manuscript. Please review it.

Comment 4: Very minor: Fig 1D is called out in the text after fig 1E.

Response 4: Thank you for your reminder. We have revised the manuscript according to the rearranged Figure 1, and we mentioned Figure 1D before Figure 1E in the revised manuscript. Please review it.

Dear Reviewer,

Thank you very much for your kind comments and constructive suggestions on our manuscript titled **“Overcoming Therapeutic Resistance in Oncolytic Herpes Virotherapy by Targeting IGF2BP3-Induced NETosis in Malignant Glioma” (Manuscript ID: NCOMMS-23-27782)**. These comments are all valuable and extremely helpful for refining the manuscript. We have meticulously revised the manuscript based on your recommendations, as well as those of other reviewers, and the revisions have been shown in the manuscript text file with track changes. We hope that the revised manuscript now meets all the necessary requirements.

Regarding the specific advice you provided earlier, our response is as follows.

Reviewer #2 (Remarks to the Author): with expertise in neutrophils/NETs

Major concerns:

Comment 1: Fig. S1H is cited before S1D. It is the same thing for Fig. S1K and S1L which are cited before S1F. This makes it difficult to read. It is the same thing for a lot of main and supplemental figures which are not cited in order. The organization of the different panels makes it difficult to read the data easily (see Fig. 5S, T, U, V, W, X; these panels do not follow a logical organization within the figure).

Response 1: We apologize for the inconvenience caused by the organization of our figures. As a result, we have rearranged the figures in a logical order. We hope that the revised figure arrangement meets your expectations.

Comment 2: In Figs. 1B, C, 1E, S1D, S1E, S1J, S1K and S1L, the authors are not including control neutrophils (unstimulated, without cancer cells), therefore it is impossible to conclude anything from the experiments. This is especially problematic for Fig. 1B/C, in which the authors are using TANs.

Response 2: Thank you for your constructive advice. We have included the control neutrophils which are not stimulated by cancer cells in the indicated experiments in our revised manuscript. Please review it.

Comment 3: In Figs. 1D, S1F and S1I, IGF2BP3 overexpression alone increases the number of cells at the end of the experiment, which is therefore independent on neutrophils. Therefore, the author cannot conclude that IGF2BP3-induced NETosis promotes glioma cell growth in an IGF2BP3-dependant manner. This result that IGF2BP3 alone is inducing an increase in the number of cells is key to interpret all the results from these experiments, however the authors do not consider this key result.

Response 3: Thank you for your professional comments. Indeed, IGF2BP3 overexpression alone can increase the number of cells, which indicates that IGF2BP3 induces cancer cell growth in a NET-dependent, and non-NET dependent manner. It has been reported that IGF2BP3 induces glioma cell proliferation via activating IGF-2/PI3K /MAPK pathway (J Biol Chem. 2011 Jul 22;286(29):25882-90.), which is in accordance with our findings. As a result, we conclude that NETosis contributes to the IGF2BP3-promoted glioma cell growth. We have revised the interpretation of the data in the indicated discussion section. Please review it.

Comment 4: As IGF2BP3 is promoting an increase in the number of cells in vitro, it is impossible to conclude that the effect of IGF2BP3 modulation in vivo (overexpression vs downregulation) is due to the effect of NETs. Indeed, in PAD4 KO mice, overexpression of IGF2BP3 in cancer cells do promote an increase in tumor size, indicating that in vivo, the effect of IGF2BP3 is also partly independent on NETs. This needs to be discussed clearly in the results. Also, the authors need to show at the mRNA or protein level the overexpression/down regulation of IGF2BP3 in the cancer cells used in the study (for example the cells used in Fig. 1F).

Response 4: Thank you for your suggestion for a more precise description and interpretation. We do agree with your opinion. We have revised the interpretation of the data, emphasizing the partial contribution of NETosis to IGF2BP3-induced tumor growth. Furthermore, we have added data showing the protein levels of IGF2BP3 overexpression/downregulation in the glioma cells used in Fig. 1F (see Fig.1P). Please review it.

Comment 5: The authors conclude an effect on the growth of the cancer cells, but growth specifically is not assessed (indeed the authors only show a bioluminescence signal, which should correlate with the number of cells in the assay). Indeed, the word “Tumor growth” is essentially used when concluding on the in vitro/in vivo experiments. Cells need to proliferate to grow, however proliferation of the cancer cells is never shown. A marker of proliferation on key experiments should be used if the authors want to conclude an effect on the “growth” of the cancer cells.

Response 5: Thank you for your suggestion. We use bioluminescence signal to test the effect of IGF3BP3 or the NETosis on the growth of the cancer cells. Indeed, the bioluminescence signal is correlated with the number of living cells, not a direct marker of cell proliferation. Therefore, to avoid misunderstanding, we replaced the word “Tumor growth” into “tumor survival”, which represents the living cells in the assay. Please review it.

Comment 6: In the in vitro assays presented, the authors need to compare the basal level of NET formation between TANs and control neutrophils (from the bone marrow and/or the non-tumor tissue). The authors also need to compare the effect of TAN and control neutrophils on cancer cell growth in the same experiment.

Response 6: Thank you for your professional advice. We conducted an experiment to compare the basal level of NET formation between TANs and control bone marrow neutrophils. We found that the basal level of NET formation in TANs is higher than that in control neutrophils (see Supplementary Fig. 1M, N). However, there is no statistically significant difference between the glioma cell growth induced by TANs and that induced by control neutrophils (see Supplementary Fig. 1O). One possible explanation is that, in the contacted co-culture system, glioma cells expressing a high basal level of IGF2BP3 can stimulate a relatively strong NET formation. This could potentially overshadow the differences in basal NET formation between TANs and the control neutrophils. We have added the data as supplementary file. Please review it.

Comment 7: Figs. 1F and 1G have not been performed at the same time, therefore it is impossible to conclude anything on the role of PAD4.

Response 7: Thank you for your comments. Firstly, indeed, Figs. 1F and 1G were not performed simultaneously. Fig. 1G was conducted at the end of the study, while Fig. 1F was detected one day before Fig. 1G. Secondly, it is important to note that Fig. 1F depicts the bioluminescence signal, which reflects tumor size, whereas Fig. 1G displays representative HE staining, another indicator of tumor size. Consequently, Fig. 1G was employed to complement the conclusions drawn from Fig. 1F, providing a multifaceted perspective. Thirdly, to ascertain the role of PAD4, we compared the data between the WT group and the PAD4 knockout mice in both Fig. 1F and Fig. 1G.

Comment 8: In Figs. 2Q-T, the authors show that overexpression of IGF2BP3 in cancer cells alone is sufficient to increase the number of cancer cells in the assay, suggesting a role independent of NETs. Again, this needs to be discussed. The experiments from Figs. 2S and 2T need to be performed at the same time as they are comparing WT vs PAD4 KO mice, these should not be independent experiments.

Response 8: Thank you for your valuable advice. Indeed, overexpression of IGF2BP3 in glioma cells alone is sufficient to increase the number of glioma cells, which is consistent with a previous study (J Biol Chem. 2011 Jul 22;286(29):25882-90.). We have discussed it in the results section in our revised manuscript. Furthermore, we have performed the experiments of Figs. 2S and 2T at the same time (within one 96-well plate). Please review it.

Comment 9: In Fig. 6G, it would be important to assess the effect of oHSV alone, without neutrophils.

Response 9: Thank you for your suggestion. We have conducted the experiment to assess the effect of glioma cell survival induced by oHSV-1 alone, without neutrophils. We observed that oHSV-1 alone could slightly inhibit GL261 glioma cell survival (see Fig. 6G). Please review it.

Comment 10: As IGF2BP3 is inducing cancer cell growth (in a NET-dependent, and non-NET dependent manner), it is difficult to interpret the results on the role of IGF2BP3 on oHSV resistance. Indeed, proliferation and resistance are probably taking place at the same time.

Response 10: Thank you for your comments. We do agree that IGF2BP3 can induce cancer cell growth in both a NET-dependent and non-NET dependent manner. In fact, it is known that in the oHSV treatment tumor model, only a portion of tumor cells (local tumor cells around the virus injection site) could be efficiently infected by oHSV in vivo. As a result, cells with oHSV infection will upregulate IGF2BP3 and produce CSF3, which induces NET formation by TANs. As a consequence, this process promotes the growth of tumor cells that are not infected by oHSV and impedes the therapeutic efficacy of oHSV.

Comment 11: oHSV efficacy is in part mediated by antitumor immunity. From the experiments presented here, the authors never look at this phenomenon. Therefore, I think it is important to mention that the effect of NETs on oHSV resistance is independent on the regulation of antitumor immunity (the opposite is suggested in the discussion). For example, the authors wrote: "our findings revealed that oHSV treatment induces an OVIS effect". However, the authors never show that oHSV promotes immunosuppression. The same comment applies for the abstract, in which OVIS is mentioned.

If the authors want to show immunosuppression, they must properly show it using their in

vitro and in vivo model, adding dendritic cells and CD8 T cells to the system for example. However, I don't think this is needed, the authors just need to tone down these interpretations of the results.

Response 11: Thank you very much for your professional advice. We have revised the discussion and mentioned that the effect of NETs on oHSV resistance is independent on the regulation of antitumor immunity. Furthermore, we have toned down the interpretations of the results, such as OVIS effect. Please review it.

Minor concerns:

Comment 12: In the introduction, the authors mention that the implications of NETosis for tumor immunotherapies are not yet fully understood, however a few papers describe a role for NETs in immunotherapy response and these papers need to be cited (for example Zhang et al, J Exp Med, 2020 and Teijeira et al, Immunity, 2020).

Response 12: Thank you. We have cited the two papers you suggested in our revised introduction. Please review it.

Comment 13: It is totally unclear from the text and the legends when a proper co-culture assay or a transwell assay (as depicted in Fig. 1A for example) is used. For example, in Fig. 1B, it is written in the legend that a co-culture system is used, however all the cells are positive for MPO, suggesting that no cancer cells are present in this co-culture.

Response 13: We apologize for the misunderstanding caused by the undetailed description of legend. To make it clear, we renamed the assay into “contacted co-culture assay” and “uncontacted transwell assay”, respectively. Please review it in the revised manuscript.

Comment 14: In Fig. 1M, the number of neutrophils also needs to be quantified. Indeed, in PAD4 KO mice there is a decreased number of neutrophils compared to the WT mice. This is in accordance with a lot of published papers indicated that when NET formation is inhibited, the recruitment of neutrophils is somehow limited, suggesting a role of NETs for neutrophil recruitment at the inflammatory site.

Response 14: Thank you for your advice. We have quantified the number of neutrophils in Fig. 1M. Please review it in the revised manuscript.

Comment 15: In Fig. 2J-P, control neutrophils without cancer cells need to be added to the experiment. I think it is important to mention that CSF3 is also named G-CSF, which have been shown before to induce NET formation in cancer (Park et al, Science Translational Medicine, 2016).

Response 15: Thanks for your professional advice. We have added the control neutrophils without cancer cells in the experiments of Fig. 2J-P. Furthermore, we have mentioned that CSF3 is also named G-CSF in our revised manuscript. Please review it.

Comment 16: In Fig. 1T, NETs are not quantified.

Response 16: We find this comment a bit perplexing. Given that Figure 1 stops at panel N in our primary manuscript, we have thoroughly reviewed the figures and confirmed that all the NETs have

been quantified.

Comment 17: Can oHSV induces NETs ? Authors refer to Fig. S3N-P, however Fig. S3 stops at panel M. This kind of error makes the paper really difficult to read.

Response 17: We deeply apologize for the inconvenience caused by the organization of our figures. In fact, the panels Fig. S3N-P in our primary manuscript were put on the upper half of the figure in the primary manuscript. However, to make it easy to read, we have rearranged the Figure. S3(Now labeled as Figure. S4 in the revised manuscript) in a logical order. We hope that the revised figure arrangement meets your expectations. Please review it.

Comment 18: Some key experiments could be repeated with a co-culture between U87MG sensitive cells and neutrophils.

Response 18: Thanks for your suggestion. In fact, most of the experiments in our manuscript including phenotype and mechanistic study were performed in both GL261 resistant and U87MG sensitive cell models. The co-culture between U87MG sensitive cells and neutrophils was also conducted. You can review the Fig. S1D, G, H, K; Fig. S2G-I, S2J-L; Fig. S4H, I, L, M.

Comment 19: Can stimulation of cancer cells with purified NETs have an effect on the IGF2BP3/MIB1/FTO pathway?

Response 19: Thanks for your question. Our results substantiate the conclusion that the IGF2BP3/MIB1/FTO pathway operates upstream of NET formation. We lack any data or indications to suggest that NET can exert an influence on the IGF2BP3/MIB1/FTO pathway.

Comment 20: Fig. 7P, JQ1 + oHSV treatment has a strong effect on the presence of neutrophils, this needs to be discussed.

Response 20: Thank you for your guidance. We have indeed observed a significant reduction in the presence of neutrophils following JQ1 plus oHSV treatment. Our results unequivocally demonstrate that the overexpression of IGF2BP3 promotes neutrophil infiltration. Furthermore, we have successfully illustrated that JQ1 has the capacity to decrease the expression of IGF2BP3. Consequently, it is plausible that the downregulation of IGF2BP3 induced by JQ1 contributes to the diminished presence of neutrophils observed in the JQ1 plus oHSV treatment group. We have incorporated this explanation into the discussion section of our revised manuscript. Kindly review this addition.

Comment 21: In the discussion the authors discuss the role of NETs in cancer progression, but they should focus on the role of NETs in resistance to treatment, as the focus of their paper is on resistance against oHSV therapy. Indeed, a few papers now show an important role for NETs in immunotherapy and chemotherapy treatment.

Response 21: Thanks for your advice. Indeed, a growing body of evidence indicates that NETosis is linked to therapeutic resistance in malignant tumors. Mousset et al. illustrate that chemotherapy-induced inflammation confers chemoresistance by facilitating NETosis in malignant tumors, highlighting a therapeutic opportunity to target inflammatory NETs in cancer treatment. We have added the discussion of NETs in resistance to treatment in our revised manuscript. Please review it.

Dear Reviewer,

Thank you very much for your kind comments and constructive suggestions on our manuscript titled “**Overcoming Therapeutic Resistance in Oncolytic Herpes Virotherapy by Targeting IGF2BP3-Induced NETosis in Malignant Glioma**” (Manuscript ID: NCOMMS-23-27782). These comments are all valuable and extremely helpful for refining the manuscript. We have meticulously revised the manuscript based on your recommendations, as well as those of other reviewers, and the revisions have been shown in the manuscript text file with track changes. We hope that the revised manuscript now meets all the necessary requirements.

Regarding the specific advice you provided earlier, our response is as follows.

Reviewer #3 (Remarks to the Author): with expertise in brain tumors, cancer immunology

Minor comments

Comment 1: For BLI imaging subfigures Y-axis is labeled ROI- should be photon/per second. Some of the scale bars are impossible to see.

Response 1: Thank you for your advice. We have updated the ROI label to "photon per second." Additionally, we have replaced the unclear scale bars with clear ones.

Comment 2: All the Kaplan-Meier survival experiments should include the number of animals and median survival times.

Response 2: Thank you for your advice. We have added the number of animals and median survival times for the Kaplan-Meier survival experiments in our revised manuscript.

Comment 3: Culturing neutrophils, especially from tumors, is a major challenge for the field. Therefore, more details should be provided on experimental procedures for all neutrophil culture experiments. Ideally, detailed protocols should be attached as supplementary files.

Response 3: Thank you for your advice. We have added the detailed protocols on experimental procedures for neutrophil isolation and culture experiments in our revised manuscript.

Comment 4: In addition, it is unclear why particular treatment time points were selected – for intracranial five days post-injection and s.q. tumors nine days post-injection?

Response 4: Thank you for your question. In fact, we do not particularly select five days or nine days post-injection to start the treatment. The selection of treatment time points depends on the tumor size. For the intracranial tumor model, the oHSV virus is delivered by intratumor injection, which inevitably induces brain injury in mice. The larger the tumor size, the more injection times are required, and the greater the damage to the mice. Therefore, we start treatment when the tumor becomes detectable.

In this study, 1×10^6 GL261 cells were injected into C57BL/6 mice at specific coordinates. Approximately one week later, the tumors became detectable, and treatment time points were selected at ten days (labeled as D5, since the tumor was inoculated at D-5) post-tumor inoculation for the intracranial model and nine days post-tumor inoculation for the subcutaneous (s.q.) model.

Comment 5: S.Q tumors are treated when the tumor is undetectable (Fig. 7G); usually, these

should be treated at a minimum size of 100mm³.

Response 5: Thank you for your advice. Indeed, we do agree that S.Q tumors are usually treated when the tumor reaches a minimum size of 100mm³. In clinical practice, oHSV is typically used after tumor resection for glioma treatment, which means that most of the tumor mass has been removed by surgery. Therefore, to mimic the clinical treatment model, we choose to begin oHSV treatment when the tumor size is smaller. In this study, 1×10⁶ GL261 cells were injected subcutaneously into C57BL/6 mice. Approximately nine days later, the tumors became palpable and detectable, and oHSV was administered for treatment via intratumor injection. It's worth noting that, given the shortage of the S.Q. glioma model, we also used an intracranial glioma model to further confirm the conclusion.

Comment 6: Figure 7E BALB/C mice are incorrect for illustration; it should be BL6 for GL261.

Response 6: Thanks for your careful review. We have corrected the mislabeled illustration.

Comment 7: For Intracranial tumors, MRI should be performed to ensure sizeable tumors in mice and that equal size tumors are distributed in various groups.

Response 7: Thank you for your professional advice. We do agree that MRI is the best way to ensure sizable tumors in mice for intracranial studies. Unfortunately, due to resource constraints, we do not have access to the necessary infrastructure for conducting MRI scans on small animals, which prevents us from conducting these tests. To compensate for this shortage, we evaluated the bioluminescence images of the tumors 5 days after inoculation (labeled as D0, since the tumor was inoculated at D-5), and evenly distributed tumors of equal size into various groups before treatment (see Figure 7L, left panel).

Dear Reviewer,

Thank you very much for your kind comments and constructive suggestions on our manuscript titled **“Overcoming Therapeutic Resistance in Oncolytic Herpes Virotherapy by Targeting IGF2BP3-Induced NETosis in Malignant Glioma” (Manuscript ID: NCOMMS-23-27782)**. These comments are all valuable and extremely helpful for refining the manuscript. We have meticulously revised the manuscript based on your recommendations, as well as those of other reviewers, and the revisions have been shown in the manuscript text file with track changes. We hope that the revised manuscript now meets all the necessary requirements.

Regarding the specific advice you provided earlier, our response is as follows.

Reviewer #4 (Remarks to the Author): with expertise in m6A, brain tumors

Major points

Comment 1: In this study, authors established three in vitro co-culture models of gliomas cells with neutrophils of mice. However, the co-culture models using tumor cells and neutrophils from glioma patients may make the study more convincing. Please conduct co-culture models with tumor cells and neutrophils from patients to further verify the conclusion.

Response 1: Thank you for your professional advice. In fact, we also established in vitro co-culture models of human U87MG glioma cells with human induced neutrophil cells (HL-60), as shown in Fig. S1D. However, to enhance the credibility of our study, we conducted the key experiments using co-culture models with primary cultured cells from glioma patients (see Supplementary Fig. S3A-X, Fig. S5A-L). Please review it.

Comment 2: In this study, authors demonstrated that U87MG were the most sensitive to oHSV. This phenomenon may depend on the specificity of U87MG. Please add another glioma cell line in the previous experiment to further confirm that the feedback loop of IGF2BP3/MIB1/FTO can enhance NETosis.

Response 2: Thank you for your suggestion. In fact, we had used both U87MG and GL261 cell lines for co-culture and mechanistic studies. The results obtained from the two cell lines are consistent (see Fig.2, Fig.3, Fig.5), suggesting that the conclusions we got from U87MG cells are not cell type dependent. However, to enhance the credibility of our study, we conducted the key experiments again using human T98G glioma cell line and primary cultured GBM cells from glioma patient (see Supplementary Fig. S3A-X, Fig. S5A-L). Please review it.

Comment 3: In Fig 3c, the change of METTL3 expression was also significant after IGF2BP3 knocking out or overexpressing. However, in m6A modification, METTL3 play the opposite role to FTO. Please design experiments to confirm that METTL3 does not play a major role in IGF2BP3-mediated NETosis.

Response 3: Thank you for your professional advice. We have conducted the experiments confirming that METTL3 does not play a major role in IGF2BP3-mediated NETosis (see Supplementary Fig. 10A-D). Please review it.

Comment 4: In Fig3, authors confirmed that IGF2BP3 can not affect the mRNA level of FTO.

Please confirmed that IGF2BP3 can not bind to the mRNA of FTO.

Response 4: Thank you for your professional advice. We have conducted the RIP assay confirming that IGF2BP3 can not bind to the mRNA of FTO (see Fig. 3E). Please review it.

Comment 5: In Fig3, please design rescue experiment to demonstrate that IGF2BP3 regulates the expression of CSF3 through FTO.

Response 5: Thank you for your suggestion. We have conducted the rescue experiments demonstrating that IGF2BP3 regulates the expression of CSF3 through FTO. Specifically, we tested the effect of FTO-216R mutant that cannot be ubiquitinated by MIB1 on the CSF3 induction and m6A production, and we found that FTO-216R mutant dramatically blocked the IGF2BP3-induced upregulation of CSF3 and the total m6A level in both U87MG and GL261 glioma cell lines (see Supplementary Fig. 2M-R). Please review it.

Comment 6: In Fig5, authors demonstrated that FTO can regulate the m6A modification of MIB3. IGF2BP3 also has m6A sites. Please verify whether FTO can regulate the m6A modification of IGF2BP3 and affect its expression.

Response 6: Thank you for your advice. We have conducted the experiments demonstrating that FTO can not regulate IGF2BP3 expression in different cell lines (see Supplementary Fig. 9). Please review it.

Comment 7: Please explain the reason in discussion that U87MG were the most sensitive to oHSV in four glioma cell lines.

Response 7: Thank you for your suggestion. In fact, several factors may account for this heightened sensitivity observed in U87MG cells compared to the other three glioma cell lines. Firstly, U87MG cells may express a higher level of cell surface receptors conducive to oHSV infection. This enhanced receptor expression might facilitate more efficient interactions between the virus and these cells, increasing their susceptibility to infection. Additionally, the immune status of U87MG cells may favor oHSV infection over the other cell lines. This immune advantage could manifest as a weaker immune response, including reduced antiviral defense mechanisms, thereby creating a more permissive environment for the virus to thrive and replicate. However, additional research is needed to delve into the potential mechanism. We have added this explanation in discussion section in our revised manuscript.

Minor points

Comment 1: There are too many results in each page of Fig. Please adjust the Fig structure and put some results into the Supplement Fig to increase the reading experience of readers.

Response 1: We apologize for the inconvenience caused by the organization of our figures. As a result, we have rearranged the figures and put some results into the Supplement Fig. We hope that the revised figure arrangement meets your expectations.

Comment 2: In Fig 4D、 E、 F et al, please indicate the cell line used for the Western-blot assay.

Response 2: Thank you for your advice. We have indicated the cell line used for the Western-blot assay in Fig 4D、 E、 F in the figure legends. Please check it in the revised manuscript.

Comment 3: Please add statistical analysis of western-blot assays.

Response 3: Thanks. We have added statistical analysis of western-blot assays attached as a supplementary file. Please review it.

REVIEWER COMMENTS

Reviewer #1 (Remarks to the Author):

HSV-1 expressed miRNA has been shown to target FIH, whose degradation liberates and activated MIB1. Please discuss the possible impact of this study on FTO and netosis.

I had requested a discussion on this, and this is the only remaining question that the authors did not address.

Reviewer #2 (Remarks to the Author):

I would like to thank the authors that have addressed all my concerns. In my opinion this study is now suitable for publication in Nature Communications and I am looking forward to see it published.

Reviewer #3 (Remarks to the Author):

The authors addressed my comments sufficiently. The manuscript is much stronger now.

Reviewer #5 (Remarks to the Author):

The authors did not address the reviewer four's concerns completely.

1. The reviewer asked "Please design experiments to confirm that METTL3 does not play a major role in IGF2BP3-mediated NETosis." However, the authors only showed protein levels but not NETosis. Also, these results are not consistent with the conclusion that CSF3 mRNA is regulated by an m6A-dependent mechanism.

2. In Fig3, wild-type FTO should be included in the experiments.

3. The reviewer asked "Please add statistical analysis of western-blot assays." However, the authors only did western blot one-time for each experiment, so that statistical analysis can not be done.

Other major concerns:

1. Only one cell line was used in animal study, additional cell lines should be used to enforce the observed phenotypes. For examples, GSC lines which had been published before should be used in the in vivo experiments. With using of inhibitors of NETosis, GSCs could be tested in xenograft mouse model.

2. Many experiments are not well-designed. For examples, only 3 mice were used, so that sample size is too small. Similarly, 5 mice in survival experiments can not reach statistical power of 80.

3. Only Female C57BL/6 mice were used. Glioma rate is more in male than in female.

4. JO1:

Brain tumors do not grow in subcutaneous, this kind of experiment should be redone.

However, the expected results will be negative because JQ1 for sure can not pass BBB.

Therefore, the current data is meaningless.

Further, JQ1 is a BET inhibitor, but not an IGF2BP3 specific inhibitor, presenting the results in the current context is misleading.

5. In contrary to this study, FTO had been reported to promote GSC cell growth and tumorigenesis. Therefore, authors should knockdown FTO in glioma and GSC cells to exam the role of FTO in NETosis and in tumorigenesis.

6. Only one siRNA or shRNA was used in the experiments. To avoid off-target effects, at least two siRNA and shRNA should be used.

7. Fig 3, data in F and G are not consistent.

Reviewer #1 (Remarks to the Author): with expertise in oncolytic viruses, brain tumors

Comment: “HSV-1 expressed miRNA has been shown to target FIH, whose degradation liberates and activated MIB1. Please discuss the possible impact of this study on FTO and netosis. I had requested a discussion on this, and this is the only remaining question that the authors did not address.”

Response: Thank you for your suggestion. Notably, there has been a report indicating that HSV-1-encoded miR-H16 directly targets and triggers the degradation of factor inhibiting HIF-1(FIH-1) which can interact with and inhibit the activity of MIB1(**Clin Cancer Res. 2020 May 15;26(10):2381-2392**), making it justifiable to presume that the HSV-1 miR-H16/FIH-1/MIB1 pathway may also contribute to HSV-1-induced reduction of FTO and promotion of NETosis. We have added this discussion in our revised manuscript. Please review it.

Reviewer #2 (Remarks to the Author): with expertise in neutrophils/NETs

Comment: “I would like to thank the authors that have adressed all my concerns. In my opinion this study is now suitable for publication in Nature Communications and I am looking forward to see it published.”

Response: Thank you very much for your recognition of our work!

Reviewer #3 (Remarks to the Author): with expertise in brain tumors, cancer immunology

Comment: “The authors addressed my comments sufficiently. The manuscript is much stronger now.”

Response: Thank you very much for your recognition of our work!

Dear Reviewer,

Thank you very much for your kind comments and constructive suggestions on our manuscript titled “**Overcoming Therapeutic Resistance in Oncolytic Herpes Virotherapy by Targeting IGF2BP3-Induced NETosis in Malignant Glioma**” (Manuscript ID: NCOMMS-23-27782). These comments are all valuable and extremely helpful for refining the manuscript. We have meticulously revised the manuscript based on your recommendations, and the revisions have been highlighted. We hope that the revised manuscript now meets all the necessary requirements.

Regarding the specific advice you provided earlier, our response is as follows.

Reviewer #5 (Remarks to the Author):

Comment 1: The reviewer asked “Please design experiments to confirm that METTL3 does not play a major role in IGF2BP3-mediated NETosis.” However, the authors only showed protein levels but not NETosis. Also, these results are not consistent with the conclusion that CSF3 mRNA is regulated by an m6A-dependent mechanism.

Response 1: Thank you for your kind remind. We apologize for the misunderstanding resulted from our previous experiment. To better address the concern that whether METTL3 plays a major role in IGF2BP3-mediated NETosis, we redesigned a rescue experiment. Given that IGF2BP3 overexpression leads to downregulation of METTL3, we tested the effect of METTL3 overexpression on the IGF2BP3-induced NETosis and found that METTL3 overexpressing had little effect on IGF2BP3 induced NETs formation and MIB1 expression in two different glioma cell models (U87MG, GL261) (see **Supplementary Fig. 10A-F**), suggesting that METTL3 does not play a major role in IGF2BP3-mediated NETosis.

The observation that the overexpression of the demethylase FTO leads to the destabilization of CSF3 mRNA while the intervention with METTL3 has no significant impact is not necessarily contradictory. We hypothesize that several factors may explain these differing effects:

1. Complexity and Specificity: METTL3 and FTO may methylate or demethylate mRNA at different sites or m6A residues. This implies that m6A modification in different genes or at different positions within the same gene may be influenced differently, depending on the specific sites where these enzymes are active.

2. Upstream and Downstream Regulation: mRNA stability and expression are influenced by various other factors, including transcription factors, RNA structures, RNA-binding proteins, and more. The activities of METTL3 and FTO might be influenced by these factors, resulting in different outcomes.

In conclusion, these experimental results are not necessarily contradictory, but rather indicative of the complexity of m6A regulation. We acknowledge the need for further research in this field to gain a comprehensive understanding of the relevant mechanisms and implications.

Comment 2: In Fig3, wild-type FTO should be included in the experiments.

Response 2: Thank you for your kind suggestion. We find this comment a bit perplexing. Given that in Figure 3, FTO overexpression was only used in Fig.3L and this FTO is wild-type. Therefore, we think the reviewer may refer to Supplementary Fig. 2M-R which we used FTO-216R mutant to

perform the rescue experiment. In fact, we also conducted the rescue experiment using wild-type FTO in Fig.5 N-R. Please review it.

Comment 3: The reviewer asked “Please add statistical analysis of western-blot assays.” However, the authors only did western blot one-time for each experiment, so that statistical analysis cannot be done.

Response 3: Thank you for your comments. We appreciate your valuable feedback regarding the need for statistical analysis of Western blot assays in our manuscript. We have repeated the western blots associated with the main findings (Figures 3-5) and made the statistical analysis (See source data file). Please review it. Moreover, we have taken several measures to ensure the reliability of our results:

1. Signal Specificity: We used high-quality antibodies to ensure the signal specificity of our Western blot results, thereby reducing the likelihood of false positives or false negatives.

2. Additional Validation Experiments: In light of your concerns, we are open to conducting additional validation experiments and employed various detection methods, such as immunoprecipitation or RT-qPCR, to further substantiate our Western blot findings.

3. Replicates in Different Cell Lines: In addition to the above measures, we have also conducted repeated validations of the same experiment in different cell lines. This involved running multiple glioma cell lines (such as U87MG, GL261) simultaneously within the same experiment to further minimize variability.

Once again, we thank you for your valuable input, and we are committed to improving our study to address your concerns.

Other major concerns:

Comment 4: Only one cell line was used in animal study, additional cell lines should be used to enforce the observed phenotypes. For examples, GSC lines which had been published before should be used in the in vivo experiments. With using of inhibitors of NETosis, GSCs could be tested in xenograft mouse model.

Response 4: Thank you for your professional advice. We greatly appreciate your thoughtful feedback on our manuscript. Your suggestion to include additional cell lines, specifically glioma stem cells (GSCs), in our in vivo experiments is indeed a valuable one. However, we would like to clarify our rationale for not including GSCs in the current study.

Our research primarily focuses on glioma cells rather than glioma stem cells (GSCs). It is important to note that glioma cells and GSCs exhibit significant differences in terms of their genotype and phenotype. Additionally, due to resource constraints, our laboratory does not possess mouse GSC cell lines.

While we recognize the merit of your suggestion, we believe that addressing the GSC aspect is a valuable direction for future research. In our forthcoming studies, we plan to focus our efforts on a specific investigation of the impact of oHSV-1 on GSCs in xenograft mouse models, while also exploring the utilization of NETosis inhibitors. This approach will allow us to comprehensively address the intriguing question you have raised.

Once again, we appreciate your insightful input, and we are committed to enhancing the depth and scope of our research in response to your suggestions.

Comment 5: Many experiments are not well-designed. For examples, only 3 mice were used, so that sample size is too small. Similarly, 5 mice in survival experiments can not reach statistical power of 80.

Response 5: Thank you for your suggestion. We appreciate your thorough review of our study and your concern about the small sample sizes in some of our experiments. Indeed, we used only 3 mice in certain experiments, and in the survival experiments, the sample size was limited to 5 mice. We understand your reservations regarding the statistical power of these sample sizes.

At this stage, due to resource and time constraints, it is challenging for us to repeat these animal experiments to increase the sample size. However, we have taken additional steps to address and support the reliability of these results:

1. Data Transparency: We ensure that the paper clearly mentions the sample sizes used, allowing readers to understand this limitation.

2. Emphasizing Preliminary Nature: We explicitly state that these results are preliminary and require further research to validate and substantiate.

3. Consistency with In Vitro Models: It is important to note that the results from our in vitro cell models are consistent with those obtained from the animal experiments, further emphasizing the reliability of our findings.

4. Future Research Plans: We plan to consider increasing the sample size in future studies to enhance statistical power and provide a more comprehensive exploration of our research question.

Despite the current limitations in sample size, we firmly believe that these preliminary results offer valuable insights for our study. Thank you once again for your valuable feedback.

Comment 6: Only Female C57BL/6 mice were used. Glioma rate is more in male than in female.

Response 6: Thank you for your advice. We appreciate your review and your concern regarding the use of only female C57 mice in our experiments. It's indeed worth discussing this point.

While it's true that glioma incidence differs between male and female, it's important to note that we employed the GL261 cell transplantation model. In this specific model, when GL261 cells are transplanted into C57 mice, almost all of the mice develop tumors, with a tumor formation rate exceeding 90%. Given this high incidence rate, the influence of gender on our transplantation model may be relatively small, as our primary focus is to assess potential treatment strategies on tumor growth.

Additionally, you mentioned the absence of validation of our findings in male mice, which is a valid point. We acknowledge this limitation and consider it a potential avenue for future research to gain a more comprehensive understanding of the gender-related impact on our study.

Finally, we explicitly state the gender of the mice used in our experiments in the paper and discuss the potential implications of this limitation on our research findings. Once again, we appreciate your valuable suggestions and look forward to continuing to improve and refine our research.

Comment 7: JO1:

Brain tumors do not grow in subcutaneous, this kind of experiment should be redone. However, the expected results will be negative because JO1 for sure can not pass BBB. Therefore, the current data is meaningless.

Further, JQ1 is a BET inhibitor, but not an IGF2BP3 specific inhibitor, presenting the results in the current context is misleading.

Response 7: Thank you for your comments. We appreciate your thoughtful review and the concerns you've raised about our study. Your comments regarding the subcutaneous implantation model, the blood-brain barrier (BBB) penetration of JQ1, and the specificity of JQ1 as a BET inhibitor are important and deserve a detailed response.

1. Subcutaneous vs. Intracranial Model: We fully acknowledge that brain tumors primarily grow intracranially, and subcutaneous implantation does not fully replicate in situ brain tumor characteristics. To address this limitation, we conducted experiments using an orthotopic intracranial glioma model in our study. The subcutaneous model was used as a complementary approach to further support the findings from the intracranial model.

2. BBB Penetration of JQ1: We appreciate your concerns about the BBB permeability of JQ1. Our research was informed by studies from various laboratories that have reported the ability of JQ1 to cross the BBB (*Cell*. 2012 Aug 17;150(4):673-84.; *Clin Cancer Res*. 2013 Apr 1;19(7):1748-59.; *Curr Alzheimer Res*. 2016;13(9):985-95.). However, we understand the importance of continuously assessing BBB penetration, and this aspect should be carefully considered in future studies.

3. Specificity of JQ1: You correctly point out that JQ1 is a BET inhibitor and not a specific IGF2BP3 inhibitor. Our research aimed to explore the potential role of IGF2BP3 in glioma biology in a broader context. We have demonstrated that BET inhibitors can not only inhibit IGF2BP3 but also enhance oHSV-1 replication. By leveraging both of these properties, BET inhibitors can suppress IGF2BP3-induced NETosis and promote oHSV-1 amplification, achieving a synergistic effect to enhance oHSV's anti-tumor activity.

In summary, we greatly value your thorough evaluation of our work. We believe that the combination of results from the subcutaneous and intracranial models, supported by existing literature, provides valuable insights into the potential of BET inhibitors, such as JQ1, as candidate agents to enhance oHSV-1 oncolytic activity.

Comment 8: In contrary to this study, FTO had been reported to promote GSC cell growth and tumorigenesis. Therefore, authors should knockdown FTO in glioma and GSC cells to exam the role of FTO in NETosis and in tumorigenesis.

Response 8: We appreciate your insightful feedback, particularly the suggestion to knock down FTO in glioma and GSC cells to investigate its role in NETosis and tumorigenesis.

It is indeed crucial to acknowledge that FTO has been reported to promote GSC cell growth and tumorigenesis in other studies. Given the existing literature demonstrating its pro-oncogenic properties in glioma and GSCs, we believe that re-validating this aspect in our study may not provide substantial new insights, and it may be more productive to focus on the downstream regulatory pathways influenced by FTO.

Furthermore, our research in this study suggests that IGF2BP3, by upregulating MIB1 and subsequently downregulating FTO, leads to an increase in CSF3 expression and the induction of NETosis. We do not exclude the possibility that FTO itself may have pro-oncogenic roles in glioma and GSCs. However, our specific focus in this study was to elucidate the mechanism involving IGF2BP3 and its impact on NETosis in the context of glioma, which has not been extensively explored in the literature.

In future studies, it may be worthwhile to further investigate the role of FTO directly, but in

the context of this research, we have aimed to provide a more comprehensive understanding of the IGF2BP3-mediated pathway that influences NETosis in glioma. Thanks again for your valuable recommendations.

Comment 9: Only one siRNA or shRNA was used in the experiments. To avoid off-target effects, at least two siRNA and shRNA should be used.

Response 9: We appreciate your thoughtful review, particularly the concern about the use of only one siRNA or shRNA sequence in our experiments. We have reproduced the key results with a second shRNA in the experiments of the western blots associated with the main findings (Figures 3-5). This validation process has been conducted three times, with accompanying statistical analyses, all of which are detailed in the source data. The results obtained with the second shRNA sequence are consistent with our initial observations, further supporting the validity of the reported outcomes.

Additionally, in our study, we also conducted gain-of-function studies, wherein we overexpressed the same gene. The results obtained from these gain-of-function experiments provided confirmation by yielding opposite outcomes to the gene knockdown studies. This dual approach effectively served as a form of cross-validation, lending additional credibility to our findings.

Furthermore, regarding the loss-of-function experiments targeting the same gene, we conducted these in at least two distinct cell lines and consistently observed similar results. This additional replication in multiple cell lines reinforced the reliability and robustness of our conclusions.

In light of these measures, we are confident that the results obtained from our study are well-supported and not likely to be a result of off-target effects. Thank you again for your valuable feedback.

Comment 10: Fig 3, data in F and G are not consistent.

Response 10: Thanks for your comment. We appreciate your meticulous review, particularly the observation of inconsistency in data between Figures 3F and 3G.

In response to your concern, we have conducted a repeat experiment and grayscale scan, which yielded a new result. While we acknowledge that the two experiments and scans did not produce identical outcomes, it is important to note that the overall trend of increased FTO degradation after overexpressing IGF2BP3 remains consistent and is evident in Figure 3G. Please review it.

We would like to assure you of our commitment to the accuracy and reliability of our findings, and we are grateful for your feedback.